# Analyzing Vision Transformers for Image Classification in Class Embedding Space

**Martina G. Vilas**
Goethe University Frankfurt
Ernst Strüngmann Institute

**Timothy Schaumlöffel**
Goethe University Frankfurt
The Hessian Center for AI

**Gemma Roig**
Goethe University Frankfurt
The Hessian Center for AI

## Abstract

Despite the growing use of transformer models in computer vision, a mechanistic understanding of these networks is still needed. This work introduces a method to reverse-engineer Vision Transformers trained to solve image classification tasks. Inspired by previous research in NLP, we demonstrate how the inner representations at any level of the hierarchy can be projected onto the learned class embedding space to uncover how these networks build categorical representations for their predictions. We use our framework to show how image tokens develop class-specific representations that depend on attention mechanisms and contextual information, and give insights on how self-attention and MLP layers differentially contribute to this categorical composition. We additionally demonstrate that this method (1) can be used to determine the parts of an image that would be important for detecting the class of interest, and (2) exhibits significant advantages over traditional linear probing approaches. Taken together, our results position our proposed framework as a powerful tool for mechanistic interpretability and explainability research.

## 1  Introduction

Transformer-based deep neural networks have become one of the most popular architectures in machine learning due to their remarkable performance and adaptability to multiple domains. Consequently, recent work has focused on reverse-engineering the inner mechanisms of these networks for better understanding and control of their predictions (e.g. [2, 7, 11, 12, 18]). Of these, a series of studies in the field of NLP [2, 7, 8, 11, 12] have shed light on the predictive mechanisms of large language transformers by projecting the hidden states and parameters of intermediate processing layers onto a vocabulary space using the pre-trained output embedding matrix. This approach has enabled the analysis in human-understandable terms of how next-word predictions are built internally, introducing a simple and efficient method for the mechanistic interpretability of NLP transformers.

So far, to the best of our knowledge, no work has shown that a similar approach can be applied to reverse engineer Vision Transformers (ViTs) for image classification. We thus introduce a method to characterize the categorical building processes of these networks by projecting their intermediate representations and parameter matrices onto the class embedding space. Our framework quantifies how the inner representations increasingly align with the class prototype learned by the model (encoded by the class projection matrix), and uncovers the factors and mechanisms behind this alignment.

In this work, we use our method to show that (1) image tokens increasingly align to class prototype representations, from early stages of the model; (2) factors such as attention mechanisms and contextual information have a role in this alignment; and (3) the categorical building process partly relies on key-value memory pair mechanisms differentially imparted by the self-attention and MLP layers. In addition, we discuss how to use this framework to identify the parts of an image that would be the most informative for building a class representation. Finally, we show that our method can

characterize the emergence of class representations in image tokens more efficiently and accurately than the commonly used linear probing approach.

## 2   Related work

Prior research has demonstrated that within ViT models, (1) the use of a linear probing approach in the hidden states of class tokens ([CLS]) in early layers enables the decoding of class representations [18], and (2) the image tokens in the final block contain class-identifiable information. By contrast, we introduce a methodological framework that adeptly extracts categorical information from image tokens early in the processing stages, and allows us to elucidate the underlying factors and mechanisms that facilitate the development of these class representations. We also demonstrate that, unlike our method, linear probes do not necessarily uncover the features relevant to the classification task.

We use our framework to complement previous findings on the inner mechanisms of ViTs. Ghiasi et al. [13] and Raghu et al. [18] investigated how tokens preserve input spatial representations across the hierarchy. We instead analyzed how tokens increasingly represent the output classes. In addition, recent work has examined and compared the role of self-attention and MLP layers: Raghu et al. [18] quantified how residual connections are differentially influenced by these sub-modules, Bhojanapalli et al. [3] measured their distinctive importance in the model's performance by running perturbation studies, and Park and Kim [17] assessed their statistical properties. We supplemented these findings by carrying out a detailed analysis of how these sub-modules build class representations by exploiting mechanisms like key-value memory pair systems. Finally, previous studies [3, 13, 16] analyzed how the performance of ViT holds against multiple nuisance factors. We alternatively inspected if class representations in image tokens are *internally* disrupted by context and attention perturbations.

Besides work probing the inner mechanisms of ViTs, tools for providing human-interpretable explanations of the network's predictions in particular instances have been developed [6]. We introduce an explainability method that follows this line of work and uncovers the parts of an image that favor the formation of meaningful class representations, independently for any block.

## 3   Interpretation of vision transformers mechanisms in class embedding space

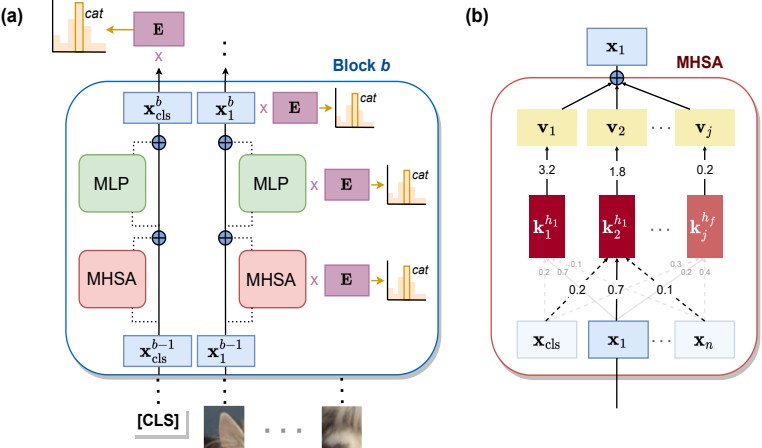

Figure 1: *Schematic of our framework.* **(a)** The hidden states of image tokens $x_n$ in a block $b$ are projected onto the class embedding space using the output embedding matrix $\mathbf{E}$. **(b)** A key-value memory pair system in a self-attention layer. Key vectors $k_j$ belong to different attention heads $h_f$. The match between the hidden states $x_n$ and the keys $k_j$ is weighted by attention values to produce a memory coefficient. Value vectors $v_j$ weighted by these memory coefficients are summed up and added to the residual stream. Adapted from Geva et al. [11].

**ViT architecture.**   As introduced by Dosovitskiy et al. [9] and depicted in Fig. 1a, vanilla image-classification ViTs take as input a sequence of linear projections of equal-sized $n$ image patches

with added position embeddings. We refer to these as image tokens $\langle \mathbf{x}_1, \ldots, \mathbf{x}_n \rangle$ with $\mathbf{x} \in \mathbb{R}^d$. The sequence also includes a special "class token" whose initial representation is learned during training, denoted [CLS]. Hence, we have $S = \langle \mathbf{x}_{cls}^0, \mathbf{x}_1^0, \ldots, \mathbf{x}_n^0 \rangle$ as the input of ViT (see bottom of Fig. 1a).

The sequence $S$ is processed by a series of transformer blocks composed of interleaving multi-head self-attention (MHSA) and MLP layers, with residual connections between each of the layers (see Fig. 1a and appendix for details). The $b^{\text{th}}$ block updates the inner representation (a.k.a. hidden state) $\mathbf{x}_i^{b-1} \in \mathbb{R}^d$ of each token $x_i \in S$ indexed by $i$, eventually producing a new hidden state $\mathbf{x}_i^b$.

Given a set of classes $C$ and a class embedding matrix $\mathbf{E} \in \mathbb{R}^{|C| \times d}$ which is learned during training, the predicted class probabilities are obtained by projecting the output of the [CLS] token of the last block and applying a softmax: $\mathbf{p}_{cls} = \text{softmax}(\mathbf{E} \cdot \mathbf{x}_{cls}^{b_n})$ (see top of Fig. 1a). We assume each row of $\mathbf{E}$ represents a *class prototype* since they encode the patterns whose detection (via matrix multiplication with the [CLS] token) determines the probability of a class being present in the image.

**Activation space projection.** To investigate how class representations emerge in ViT, we analyzed the alignment between the intermediate representations of the tokens with the class prototypes encoded by the final projection matrix $\mathbf{E}$. The key insight is that we have $\mathbf{x}_i^b \in \mathbb{R}^d$. Hence, we can obtain the prediction of the class distribution $p_i^b \in \mathbb{R}^{|C|}$ for the intermediate representation of the $i^{\text{th}}$ token in the output of the $b^{\text{th}}$ block by computing $\mathbf{p}_i^b = \mathbf{E} \cdot \mathbf{x}_i^b$ (see Fig. 1a).

To quantify the alignment, we take inspiration from Brunner et al. [4] and adapt a measure of *identifiability*. Concretely, we evaluate how recoverable the correct class $c_j$ of the $j^{\text{th}}$ image is from the class projection of the $i^{\text{th}}$ token using a measure of their *class identifiability*, $r_i^j$, that we define as:

$$r_i^j = 1 - \frac{argwhere(argsort(\mathbf{p_i}) = c_j)}{|C|} \tag{1}$$

where *argsort* sorts the logits assigned to each class from higher to lower, and *argwhere* returns the index of the correct class in the sorted vector. We normalize and reverse $r_i^j$ to obtain a score that ranges from 0 to 1, with 1 denoting that the correct class has the highest logits and 0 the lowest.

*Comparison with NLP research*: The idea of projecting the intermediate token representations onto the class embedding space to uncover the inner mechanisms of ViT is based on previous work in NLP that takes a similar approach with autoregressive language transformers [2, 7, 8, 11, 12]. These models, trained to predict the next word of a sentence, include an output embedding matrix that translates the final hidden states of the network to a human-interpretable vocabulary space. Taking advantage of this component, Geva et al. [11] and [12] projected the outputs of intermediate MLP layers onto the vocabulary space to show that these sub-modules can be decomposed into a set of sub-mechanisms promoting human-interpretable concepts.

In comparison to NLP models, ViTs are trained to predict the presence in an image of a more restricted set of classes. Therefore, the semantic insights obtained with the output embedding projection differ: in the case of NLP we can uncover linguistic anticipatory mechanisms, while in the case of ViT we can investigate how categorical representations are built.

**Parameter space projection and key-value memory pairs.** Previous work [7, 11, 12] has demonstrated that the learned parameters of transformer architectures can also be projected onto the output embedding space for reverse engineering purposes. These studies further propose that the parameter matrices can be interpreted as systems implementing key-value memory pair mechanisms, to better understand the mappings between the inputs of a model and its predictions.

As shown in Fig. 1b, key-value memories are a set of $j$ paired vectors $M = \{(\mathbf{k}_1, \mathbf{v}_1), \ldots, (\mathbf{k}_j, \mathbf{v}_j)\}$ where the keys $\mathbf{k}_i$ are used to quantify the presence of a set of patterns in the input, and the values $\mathbf{v}_i$ represent how the input should be modified in response to the detection of such pattern. We next explain how this system could be implemented in the MLP and self-attention layers of ViT.

An MLP layer in ViT consists of:

$$\text{MLP}(\mathbf{X}) = \text{GELU}(\mathbf{X}\mathbf{W}_{\text{inp}})\mathbf{W}_{\text{out}} \tag{2}$$

where $\mathbf{X} \in \mathbb{R}^{n \times d}$ represents the $n$-token input, each token of dimension $d$, and $\mathbf{W}_{\text{inp}} \in \mathbb{R}^{d \times |M|}$ and $\mathbf{W}_{\text{out}} \in \mathbb{R}^{|M| \times d}$ represent the parameter matrices.

The main idea is that the columns of $\mathbf{W}_{\text{inp}}$ and the rows of $\mathbf{W}_{\text{out}}$ can be thought of as a set of key-value paired vectors, respectively. The result of the operation shown in blue in eq. 2 is a matrix of memory coefficients. Entry $i, j$ in the matrix contains the coefficient resulting from the dot product of token $i$ with key $j$. This matrix is the dynamic component of the system and measures the presence of certain patterns in the hidden state. The matrix shown in red in eq. 2 (here we think of them as value vectors) encodes how the internal representations $\mathbf{x}_i$ should change in response to the detection of such patterns. To modify the hidden states of the network, the weighted value vectors of all key-value memories are summed up and added to the residual (see top of Fig. 1b).

In the case of the self-attention layers, the decomposition of the layer into keys and values is more complex. The core idea is as follows (see Fig. 1b). In transforming the hidden representation $\mathbf{x}_i$ (row $i$ in $\mathbf{X}$), the self-attention layers can be thought of as implementing a system of key-value memory pairs where the keys not only detect the presence of a certain pattern in $\mathbf{x}_i$ but also in the hidden states of all other tokens $\{\mathbf{x}_j : j \neq i\}$ in the sequence $S$. The coefficient reflecting the match between a key and a token $\mathbf{x}_i$ is weighted by the attention values between $\mathbf{x}_i$ and all tokens in a self-attention head (including itself). Formally,

$$\text{MHSA}(\mathbf{X}) = \text{hconcat} \left[ \mathbf{A}^1 \mathbf{X} \mathbf{W}^1_{V_{\text{attn}}}, \dots, \mathbf{A}^f \mathbf{X} \mathbf{W}^f_{V_{\text{attn}}}, \right] \mathbf{W}_{\text{out}} \tag{3}$$

where $\mathbf{A}^h \in \mathbb{R}^{n \times n}$ are the attention weights of the $h^{\text{th}}$ head with $f$ being the number of heads, and $\mathbf{W}^h_{V_{\text{attn}}} \in \mathbb{R}^{d \times \frac{d}{f}}$. The result of $\mathbf{A}^h \mathbf{X} \mathbf{W}^h_{V_{\text{attn}}}$ for every $h^{\text{th}}$ head is concatenated horizontally. The output of the horizontal concatenation is a matrix of dimensions $n \times |M|$ with $|M| = d$, which is then multiplied with the matrix $\mathbf{W}_{\text{out}} \in \mathbb{R}^{|M| \times d}$ containing the value vectors. Of note, we can say that the matrices $\mathbf{W}^h_{V_{\text{attn}}}$ of every $h^{\text{th}}$ attention head represent the key vectors of the system.

*Comparison with NLP research*: Autoregressive language transformers employ an input embedding matrix to convert a sequence of semantically meaningful elements within a vocabulary space into a sequence of machine-interpretable hidden states. These hidden states are subsequently processed by a series of transformer blocks, and the resulting output is used for prediction by projecting it back to the vocabulary space using an output embedding matrix. In many architectures, a single vocabulary-projection matrix functions as both the input and output embedding of the model, with the output embedding matrix essentially being the transpose of the input embedding matrix. Consequently, previous work in NLP examined how the keys and values of the memory system represent patterns that are translatable to a human-interpretable vocabulary [11, 12]. For example, in NLP transformers some keys in MLP layers have been reported to detect thematic patterns, such as a reference to TV shows [11]. In turn, the detection of a TV show "theme" was associated with a value vector that promotes related concepts in the vocabulary space (e.g. "episode", "season").

Unlike NLP transformer models, in ViT the mappings of the input embedding matrix (projecting from image patches to hidden states) differ from those of the output matrix (projecting from hidden states to class representations). Therefore, the keys of ViT may represent patterns that are interpretable in the image input space, while the value vectors may represent updates interpretable in the class embedding space. Furthermore, interpreting the keys of ViT is not as straightforward as in the NLP case. In autoregressive language transformers, the input space retains significance throughout the network's hierarchy since it aligns with the output prediction space. In ViTs, however, the input space is not relevant for later projections. Given that our work aims to understand how categorical representations are formed, we focus on analyzing the representations of the value vectors. We leave to future work the analysis of what the keys encode in human-interpretable terms.

## 4   Experimental design

**Vision Transformer models.**   We validate our approach using a variety of ViTs that differ in their patch size, layer depth, training dataset, and use of architectural modifications. [1] Specifically, we separately probed: 1) a vanilla ViT with 12 blocks and a patch size of 16 pre-trained using the ImageNet-21k dataset and fine-tuned on ImageNet-1k (ViT-B/16) [9]; 2) a variant with a bigger patch size of 32 (ViT-B/32) [9]; 3) a deeper version with 24 blocks (ViT-L/16) [9]; 4) a variant fine-tuned on the CIFAR100 dataset [14]; 5) a version pre-trained on an alternative Imagenet-21K dataset with higher-quality semantic labels (*MIIL*) [20]; 6) a modified architecture with a refinement module

---

[1]Our code is available at `https://github.com/martinagvilas/vit-cls_emb`

that aligns the intermediate representations of all tokens to class space (*Refinement*) [15]; and 7) an alternate version trained with Global Average Pooling (GAP) instead of the [CLS] token.

**Image dataset.**   For conducting our studies we used the ImageNet-S dataset [10], which consists of a sub-selection of images from ImageNet accompanied by semantic segmentation annotations. We analyzed the representations of 5 randomly sampled images of every class from the validation set.

# 5   Class representations in image tokens across the hierarchy

During ViT's pre-training for image classification, the only token projected onto the class-embedding space is the [CLS] token from the last block. Thus, whether image tokens across the hierarchy can be translated to the class-embedding space remains an open question. In this section, we provide an affirmative answer.

**Class representations in image tokens.**   First, to establish whether the hidden representations of image tokens encode class-specific representations, we projected the hidden states of the image tokens $\mathbf{X}$ from the last block onto the class-embedding space using the output embedding matrix $\mathbf{E}$, by computing $\mathbf{E} \cdot \mathbf{X}^T$. We then measured the model's *class identifiability rate*, which we quantified as the percentage of image tokens that contain a class identifiability score of 1. Similarly to Ghiasi et al. [13], we found that the identifiability rate of all ViTs was significantly higher than chance, and the rate was further influenced by the variant probed (see Table 1). We additionally measured the percentage of images that contain at least one image token with an identifiability score of 1 in the last block. We found that, for all variants tested on ImageNet-S, the percentage was significantly higher than in a model initialized with random weights, and higher than their corresponding top-1 accuracy scores in the classification task (see appendix). The latter finding implies that even misclassified samples retain information within some of their image tokens that correspond to the correct class.

Table 1: *Class identifiability rate (%) of image tokens in the last block.*

| ViT-B/32 | ViT-B/16 | ViT-L/16 | MIIL | CIFAR100 | Refinement | GAP |
|----------|----------|----------|-------|----------|------------|-------|
| 60.73 | 67.04 | 72.58 | 78.52 | 90.35 | 79.64 | 52.09 |

**Evolution of class representations across the hierarchy.**   Second, to investigate if class prototype representations can be decoded from tokens at various stages of ViT's hierarchy, we projected the hidden states of every block onto the class embedding space and examined the evolution of their class identifiability scores. Fig. 2 shows that the mean class identifiability scores of both image and [CLS] tokens increased over blocks, for all ViT variants. Additionally, the results demonstrate that image tokens exhibited greater variability in their scores in comparison to [CLS] tokens (see variant-specific plots in the appendix). This observation suggests that the development of class representations varies among image tokens.

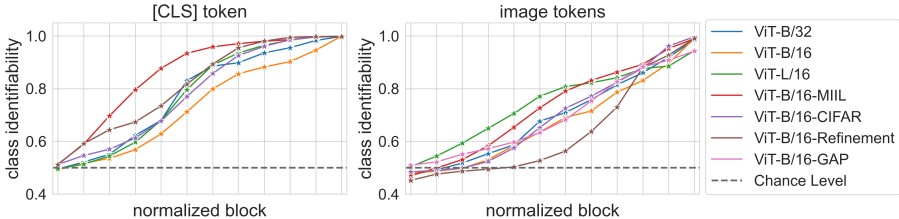

Figure 2: *Evolution of class identifiability mean scores across normalized blocks.*

After establishing that image tokens can be effectively translated to the class embedding space across different levels of the hierarchy, we proceed to showcase the applicability of our framework in examining some underlying factors contributing to the development of class representations. We exemplify this process using the simplest model variant, ViT-B/32.

**The impact of attention mechanisms on the class representations of image tokens.** To investigate whether image tokens need to incorporate information from other tokens via attention mechanisms to build a class-identifiable representation, we conducted perturbation experiments. For the hidden representation of every token $i$ we set to 0 all attention weights from each image token $\mathbf{x}_i$ to every other image token $\{\mathbf{x}_j : j \neq i\}$. We found that the perturbations erased the class identifiability of image tokens in the last block (mean class identifiability decreased from 60.73% to 0.12%). The results suggest that image tokens cannot develop class-identifiable representations in isolation.

In contrast, when removing the attention weights between the image tokens and the [CLS] token, the class identifiability rate of image tokens remained unchanged, implying that image tokens do not need to extract information from the [CLS] token to build a class representation. These results are aligned to those of Zhai et al. [21], who showed that ViTs trained without the [CLS] token can achieve similar performance to models that include it.

**The impact of context on the class representations of image tokens.** Class-identifiable representations may emerge earlier and more strongly in image tokens that belong to portions of an image depicting the class (e.g., striped patterns in an image of a zebra). Leveraging the semantic segmentation labels from the ImageNet-S, we compared the identifiability rate of class-labeled and context-labeled image tokens. Our results confirmed our hypothesis, revealing earlier and more identifiable class representations in class- than context-labeled tokens (see Fig. 3a for ViT-B/32 and appendix for the other variants). However, in deeper blocks, both types of tokens exhibited high identifiability scores, indicating that class representations also emerged in image tokens that do not depict the class. This might be the result of context tokens extracting class-relevant information via attention mechanisms from class-labeled tokens, or it might stem from the class prototype representation incorporating information about the context commonly associated with the class.

To further explore the extent to which class-labeled tokens are needed for context-labeled tokens to build a class representation, and vice-versa, we conducted a token-perturbation study in ViT-B/32. We removed either class- or context-labeled tokens from the input (after the addition of position embeddings), and measured the class-identifiability rates of the remaining image tokens in the last block. We found that in both cases the removal of information reduced to a certain extent the class identifiability scores. Concretely, the original identifiability rate of class-labeled tokens of 71.91% decreased to 44.70%, while that of context-labeled tokens decreased from 56.24% to 38.68%. On the one hand, these results suggest that class-labeled tokens need context for building more identifiable class representations. On the other hand, they show that context-labeled tokens can build class-identifiable representations without the classes themselves being depicted in the image, which suggests that ViTs have incorporated into their class prototypes contextual information. This latter finding is in line with previous studies showing that CNNs can identify a class based on context pixels only [5], leading to impoverished performance in out-of-distribution scenarios.

## 6 Mechanistic interpretability applications

After establishing our ability to project ViT's internal representations onto the class embedding space to investigate the development of categorical representations, this section elaborates on how this framework can examine the involvement of self-attention and MLP layers in this process. Our findings indicate that both types of layers contribute to the building of class representations through key-value memory pair mechanisms. MLP layers leverage this system to produce strong categorical updates in late blocks that are highly predictive of the model's performance, while self-attention layers promote weaker yet more disseminated and compositional updates applied earlier in the hierarchy.

**Building of class representations.** To investigate the extent to which self-attention and MLP layers help build categorical representations, we measured the *class similarity change rate* induced by these sub-modules. Given a layer with input/output tokens and a class embedding matrix, we computed the proportion of output tokens whose correct class logits increase relative to the corresponding logits for input tokens, where all logits are obtained by projecting the tokens onto the class embedding space (see Fig. 1a). Concretely, we projected the output of each layer $\mathbf{O}^l(\mathbf{X})$ onto the class embedding space by $\mathbf{p}_{\text{out}} = \mathbf{E} \cdot \mathbf{O}^l(\mathbf{X})^T$, and compared it to the projection of the input itself $\mathbf{p}_{\text{inp}} = \mathbf{E} \cdot \mathbf{X}^T$. We then quantified the proportion of tokens $i$ where $p_{\text{out}}^i > p_{\text{inp}}^i$.

We found that, although the evolution of the class similarity change rate differed across ViT variants (see appendix and Fig. 3b for ViT-B/32), some general patterns can be observed. First, self-attention layers exhibited a higher-than-chance class similarity change rate during at least some processing stages. The increment was generally higher for the [CLS] tokens, which trivially need attention mechanisms for creating a class representation given that their initial input is identical across images. In contrast, the class similarity change of MLP layers peaked in the penultimate(ish) blocks (except for the GAP variant) but was otherwise below chance level. The next sections investigate the reasons behind these peaks.

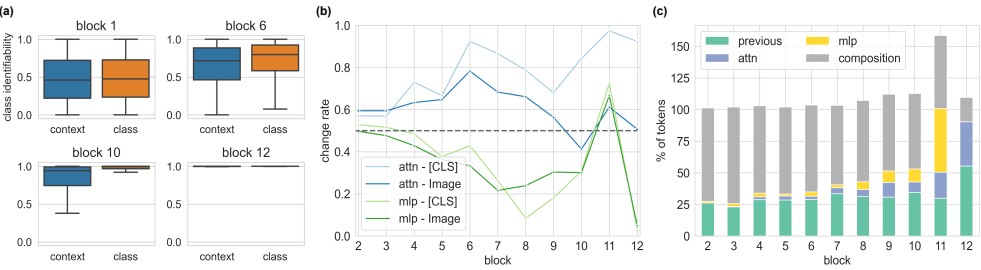

Figure 3: *Results for ViT-B/32.* **(a)** Class identifiability scores of class- and context-labeled tokens; **(b)** Class similarity change rate induced by self-attention and MLP layers; **(c)** Match between the top-1 predictions of the layers and the top-1 predictions of the residual stream.

An open question from the previous experiment is how the class representations developed by the self-attention and MLP layers are incorporated into the residual stream. To measure this, we took inspiration from Geva et al. [11] and quantified the proportion of tokens whose top-1 prediction in the residual stream equaled that of the self-attention layer, that of the MLP layer, that of the residual stream before being processed by these layers, or neither of them (indicating that the residual stream has built a new composite prediction).

We found that in early blocks the predictions of the residual stream were mostly compositional, integrating information from the self-attention and MLP layers but not being dominated by their outputs (see appendix and Fig. 3c for ViT-B/32). At later stages, the residuals incorporated more directly the predictions of the sub-modules. Results showed that the influence of self-attention layers peaked in the last block. In the MLP case, the highest values were found in block 11 (or in layers closer to the last block in the case of ViT-L/16). The exception of this latter pattern was ViT-B/16 trained with GAP, where MLP influenced the residual of the last block the most. This could be due to being the only variant where the final MLP layer can influence the classification output. In the other models, the previous to last MLP layers are those from which the [CLS] token can ultimately extract information for classification (through the self-attention layer in the last block).

**Categorical updates.** To investigate how self-attention and MLP layers carry out categorical updates by promoting class-prototype representations, we projected their output parameter matrices ($\mathbf{W}_{\text{out}}$ of eq. 2 and eq. 3) onto the class-embedding space, by computing $\mathbf{P}_{W_{\text{out}}} = \mathbf{E} \cdot \mathbf{W}_{\text{out}}^T$. The rows of $\mathbf{E}$ and $\mathbf{W}_{\text{out}}$ were normalized to unit length to enable comparison across ViT variants. This projection measures the extent to which each row in the output parameter matrices reflects the classes encoded in the embedding space, and thus probes whether the value vectors of the key-value memory pair system have high correspondence with class prototype representations. We were interested in evaluating the extent to which class prototypes had at least one memory value in a given layer $l$ that resembles their representation. Hence, we extracted the maximum value of each row in $\mathbf{P}_{W_{\text{out}}}^l$ that denotes the highest similarity score obtained per class prototype in a given layer $l$. We call this measure *class-value agreement score*.

We found that, in deeper blocks, the class-value agreement scores of self-attention and MLP layers were significantly higher than those of a model initialized with random weights (Fig. 4a and appendix). This could be due to the model developing more complex and semantically meaningful keys at deeper stages. Results also showed that the peaks in MLP layers were higher than those of self-attention layers, indicating that MLP sub-modules promote stronger categorical updates. In addition, a manual inspection of the MLP layers containing high class-value agreement scores revealed that the value

vectors may promote and cluster semantically similar and human-interpretable concepts (see appendix for examples). These patterns were observed across all variants except for ViT-GAP. Given that the later model averages the representations of the image tokens for the classification task, we hypothesize that in ViT-GAP categorical representations might be distributed across value vectors instead.

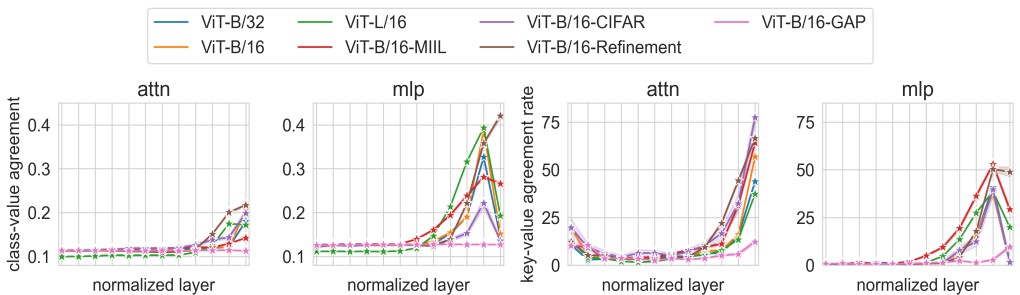

Figure 4: *Key-value memory pair mechanisms.* **(a)** Class-value agreement scores, which measure the extent to which the layers promote class prototype representations; **(b)** Key-value agreement rates, which quantify the proportion of tokens promoting the correct class using key-value memory pair systems.

**Key-value memory pairs at inference time.** To investigate if self-attention and MLP layers act as key-value memory pair systems at inference time, we measured how the keys that are most activated during the processing of a sample correspond to the value vectors that promote the representation of the correct class. Concretely, for each layer, we quantified the proportion of tokens where the 5 keys with the highest coefficients are associated with value vectors whose top-5 logits (obtained by the projection $\mathbf{E} \cdot \mathbf{W}_{\text{out}}$) indexed the correct class. We call this metric *key-value agreement rate*.

As shown in Fig. 4b, in earlier blocks, the key-value agreement rates of self-attention layers were higher than those of MLP layers. The agreement rate of MLP layers peaked in the penultimate blocks, while self-attention layers peaked in the last blocks. The rates on these peaks were generally higher for self-attention layers. These findings indicate that, while self-attention layers promote weaker categorical representations (see previous section), the updates are carried out in more image tokens than in MLP layers. Of note, similarly to the class-value agreement score results, ViT-GAP exhibited a limited use of key-value memory pair systems.

We further evaluated the influence of these mechanisms on the accuracy of the model, and compared the agreement rate of correctly classified vs. misclassified samples. We found that, for all variants excluding ViT-GAP, the agreement rates in the penultimate MLP layers of correctly classified samples were significantly higher (see appendix). In addition, in the majority of ViTs, a smaller yet significant difference was also found in the deeper self-attention layers. These findings suggest that the use of key-value memory pair mechanisms has a meaningful effect on the performance of the model.

**Compositionality of key-value memory pair mechanisms.** The final output of a layer may combine the prediction of many key-value memory pairs, and predict a class distribution that is different from that of its most activating memories. Inspired by Geva et al. [11], we measured the compositionality of the key-value memory pair system by quantifying the number of instances where a layer's final predictions matched any of the predictions of the top-5 most activated memories for that instance.

Results showed that for all variants except ViT-GAP, the penultimate MLP layers have low compositionality scores: between 50% and 70% of instances had a final prediction that matched the prediction of one of the most activated memories (see appendix). Self-attention layers also decreased their compositionality in the last blocks, but their scores were still higher than MLP layers: more than 70% of instances had a composite prediction (except for ViT-CIFAR, see appendix).

## 7 Explainability application

In this section, we show that our framework can also be used to identify the parts of an image that would be the most important for detecting a class of interest, at each processing stage of ViT.

Having demonstrated that class representations develop gradually over the ViT hierarchy, and that image tokens differentially represent categorical information, we propose to use a gradient approach to quantify how much an image token of a given block would contribute to form a categorical representation in the [CLS] token. In detail, our explainability method can be applied as follows:

1. For the $j^{\text{th}}$ image and the $b^{\text{th}}$ block, project the hidden representation of the [CLS] token after the attention layer $\mathbf{x}_{\text{cls}}^{b_{\text{attn}}}$ onto the class embedding space, and compute the cross-entropy loss $L_{\text{CE}}$ of the class of interest $c_j$, such that $\ell_j^b = L_{\text{CE}}(\mathbf{E} \cdot \mathbf{x}_{\text{cls}}^{b_{\text{attn}}}, c_j)$. This projection quantifies how close is the representation of the [CLS] token to the prototype representation of the class of interest.

2. Compute the gradient of $\ell_j^b$ with respect to the attention weights $\mathbf{a}_j^b$ that the [CLS] tokens assigned to the image tokens in an attention head in the self-attention layer, such that $\nabla \ell_j^b = -\partial \ell_j^b / \partial \mathbf{a}_j^b$. Since we are interested in how the image tokens decrease the cross-entropy loss, we negate the gradients. In simple terms, this step estimates the rate at which an image token would increase the correct class representation of the [CLS] token if more attention were allocated to it.

The final output will consist of importance scores assigned to every image token of a block and attention head that can be visualized as a heatmap over the image (see Fig. 5a). Of note, the block- and attention-head-specific visualizations of our explainability framework differ from other widely used methods that generate global relevancy maps, for example by computing the gradients of the class logits at the final layer with respect to the input and/or aggregating the information propagated across layers up to the class embedding projection (see [6] for an overview). Instead, our method can visualize the categorical information contained in the image tokens independently for each block and attention head. This allows us to (1) better understand how categorical information is hierarchically built, and (2) characterize the importance of each block in building the class representations.

In addition, our method is class-generalizable and can be used to identify the features that would lead to the assignment of different class labels (see Fig. 5). Thus, it can be used to uncover the parts of an image that produce an incorrect prediction or trigger different predictions in a multi-label classification task. Moreover, since our framework identifies the image tokens that would increase the categorical representations of the [CLS] token, its results can be compared to the actual attention weights assigned by this type of token, to shed light on what aspects of an image the network neglects or emphasizes in relation to the class-optimal ones.

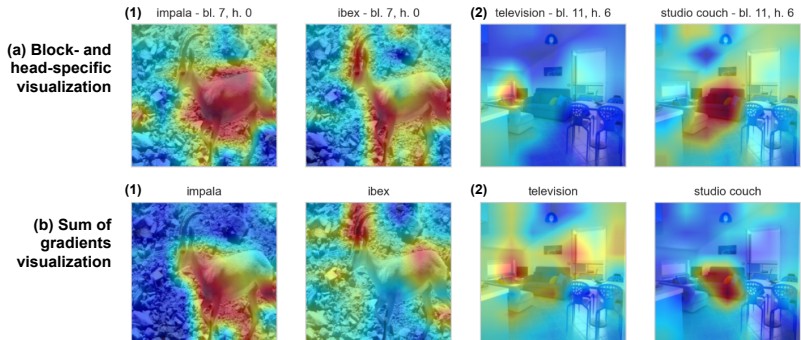

Figure 5: *Examples of feature importance visualization in ViT-B/32.* **(1)** Image example where the correct label is "Impala", but the model predicts "Ibex". **(2)** Image example with multiple labels.

Besides providing block- and attention-head-specific visualizations, our framework can also be used to generate a traditional global relevancy map. Concretely, we can aggregate the gradients over the blocks and attention heads, by $\sum_b \nabla \ell^b$, and obtain a final feature importance map that takes into account the relevance of each image token at every stage of processing (see Fig. 5b). The sum procedure also allows us to corroborate that we obtain a fair portrayal of the individual contribution of each block and attention head to the final class representation.

To validate the quality of our global relevancy maps, we compared our approach with an established explainability method [6]. Testing ViT-B/32, we separately applied both methods and: (1) Quantified

the importance of each image token; (2) Gradually removed the tokens with the least to most importance scores (negative perturbation test); (3) Gradually removed the tokens with the most to least importance (positive perturbation test); (4) Measured the accuracy of the model with each removal; (5) Computed the Area Under the Curve (AUC) of the final accuracies. Our results showed that our framework yields similar results to those of Chefer et al. [6] (see appendix), highlighting the adequacy of our approach. In addition, we found that we could maintain, and even improve, the accuracy of ViT-B/32 with the removal of up to 60% of the least important tokens. More generally, the AUC of our perturbed model was significantly higher (neg. perturbation) and lower (pos. perturbation) than that of a model whose token removal order was randomly determined.

## 8    Comparison with a linear probing approach

Linear probing is a widely used interpretability tool that consists of training linear classifiers to uncover the features represented in the inner layers of a model. Similarly to the goals of our interpretability framework, to shed light on the relevant factors behind category-building processes in neural networks, linear probing is sometimes used to predict from internal layers the classes represented in the output projection matrix [1]. However, previous studies (e.g. [19]) have shown that this approach can highlight features that are not relevant to the model's task. In other words, a successful probing result can reflect the separability of feature representations that are ignored by the model in the class embedding space. In contrast, our framework directly quantifies how the inner representations of ViT increasingly align with the representations encoded in the class prototypes.

To substantiate the disparity in insights derived from both methods, we conducted the following experiment. Reproducing the linear probing approach taken by Raghu et al. [18], we trained separate 10-shot linear classifiers on ImageNet-S for each token position and layer of a ViT-B/32. To test if the information learned with these probes sheds light on the categorical decisions taken by the network, we conducted negative and positive perturbation tests. Concretely, we quantified the class identifiability scores obtained from the linear probes for each token, gradually removed those with the least to most identifiable scores (for negative perturbation; vice-versa for positive perturbation), and measured the accuracy of the model with each removal. We compared these results with those of our framework's positive and negative perturbation experiments reported in the previous section. We found that even if linear probes could generally decode with better top-1 accuracy the classes in some of the inner layers, the obtained scores do not significantly predict the relevance of the image tokens in the categorical decision of the network (see appendix).

Notably, when used for the same goals, our framework (1) is more time-efficient: it comprises a one-forward pass on validation images, while linear probes additionally involve a one-forward pass over the training images and the fitting of a linear classifier for every token position and layer; and (2) can be used in low-resource scenarios, where the training of additional models is difficult due to small datasets or reduced computing capabilities.

## 9    Conclusions

With the increasing use of ViTs in the field of computer vision, it is necessary to build methods to understand their inner mechanisms and explain their predictions. Our work introduces an intuitive framework for such purposes that does not require optimization and provides interesting, human-interpretable insights into these networks. Concretely, we demonstrated how our method can extract class representations from the hidden states of image tokens across the hierarchy of ViT, providing insights into the category-building processes within these networks. Additionally, we used our framework to elucidate the distinct roles of self-attention and MLP layers in this process, revealing that they promote differential categorical updates that partly depend on key-value memory pair mechanisms. Lastly, we emphasized the utility of this method for explainability purposes, aiding in the identification of the most pertinent parts of an image for the detection of a class of interest.

**Limitations.**   This work only studies ViTs trained for image classification. Future work could investigate how to adapt our framework to examine the inner representations of models with other types of output embeddings. In addition, we did not explore how our approach might be used for model editing or performance improvement purposes. Some mechanistic interpretability insights gained in this work point to aspects of ViT that could be manipulated for these goals in future studies.

## 10 Acknowledgments

This project was partly funded by the German Research Foundation (DFG) - DFG Research Unit FOR 5368. We are grateful to access to the computing facilities of the Center for Scientific Computing at Goethe University, and of the Ernst Strüngmann Institute for Neuroscience.

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
