# Appendix

## 1  Interpretation of vision transformers mechanisms in class embedding space

### 1.1  ViT architecture and hidden state projection

The following update equations capture the action of the $b^{\text{th}}$ block on the $i^{\text{th}}$ token's hidden representation $\mathbf{x}$:

$$\mathbf{o}_i^{a_b} = \text{MHSA}^b(\mathbf{x}_i^{b-1}) \tag{1}$$

$$\mathbf{x}'^b_i = \mathbf{o}_i^{a_b} + \mathbf{x}_i^{b-1} \tag{2}$$

$$\mathbf{o}_i^{m_b} = \text{MLP}^b(\mathbf{x}'^b_i) \tag{3}$$

$$\mathbf{x}_i^b = \mathbf{o}_i^{m_b} + \mathbf{x}'^b_i \tag{4}$$

where MHSA is a multi-head self-attention layer and MLP is a multi-layer perceptron layer, whose outputs $\mathbf{o}$ are denoted by an upperscript $a$ and $m$, respectively.

For a given block $b$ and token $i$, we can obtain the class prediction $\mathbf{p}_l = \mathbf{E} \cdot \mathbf{z}_l$ based on the output $\mathbf{z}_l \in \{\mathbf{o}_i^{a_b}, \mathbf{o}_i^{m_b}, \mathbf{x}_i^b\}$ of each layer. We do this for all combinations of blocks and tokens.

# 2 Class representations in image tokens across the hierarchy

## 2.1 Class identifiability in image tokens

Table 1: Percentage of images in the Imagenet-S validation set with at least one image token with an identifiability score of 1 (*Top-1 CI*), and the top-1 classification accuracy (*Top-1 acc*).

|           | ViT-B/32 | ViT-B/16 | ViT-L/16 | MIIL  | Refinement | GAP   |
|-----------|----------|----------|----------|-------|------------|-------|
| **Top-1 CI**  | 95.58    | 97.45    | 97.12    | 95.52 | 96.26      | 98.53 |
| **Top-1 acc** | 81.71    | 85.69    | 86.00    | 86.26 | 84.97      | 84.13 |

## 2.2 Class identifiability evolution

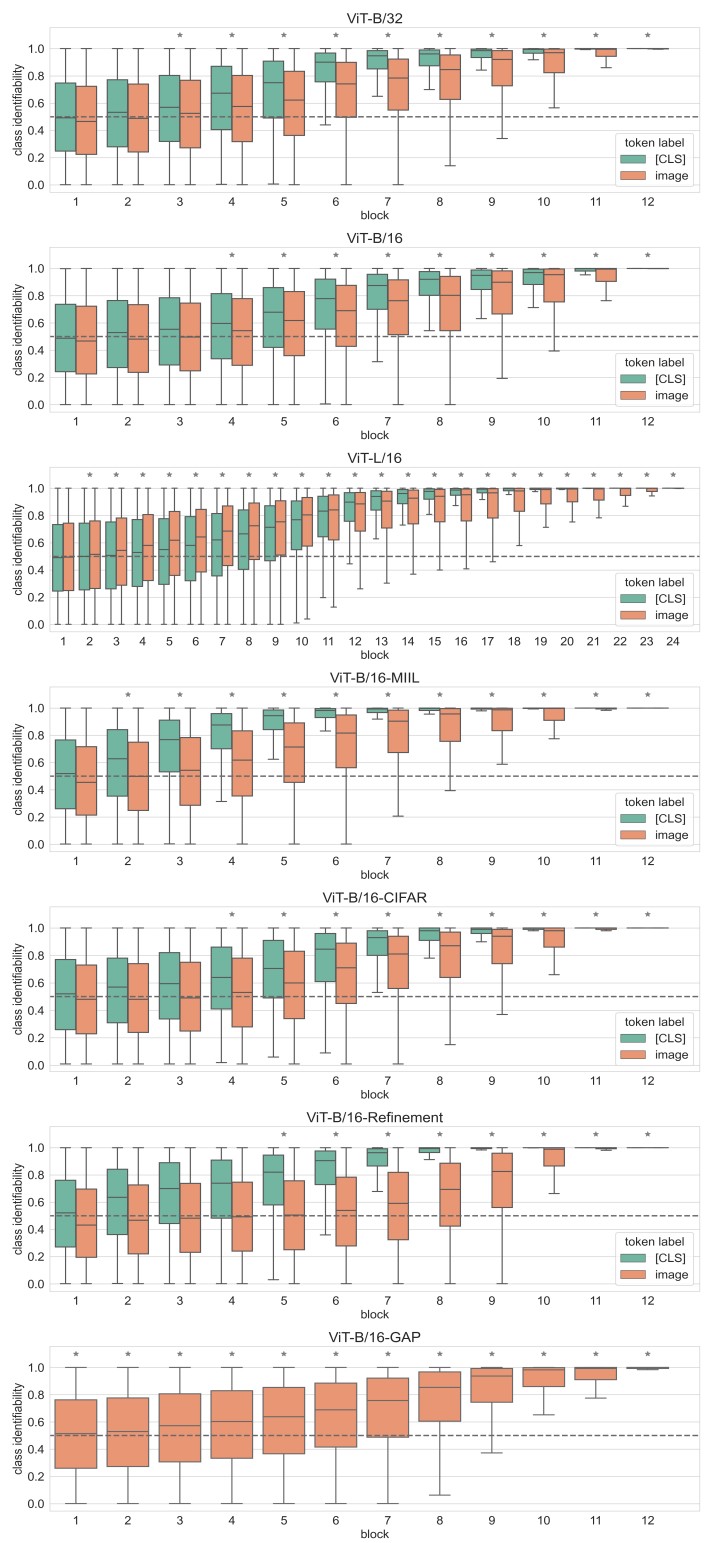

Figure 1: Class identifiability evolution of all ViT variants. Asterisks indicate blocks where the class identifiability scores were higher than those of a randomly initialized model.

## 2.3   Class similarity change rate of blocks

We additionally computed the class similarity change rate of each block. Concretely, we computed the percentage of image tokens that increment the logits of the correct class per block.

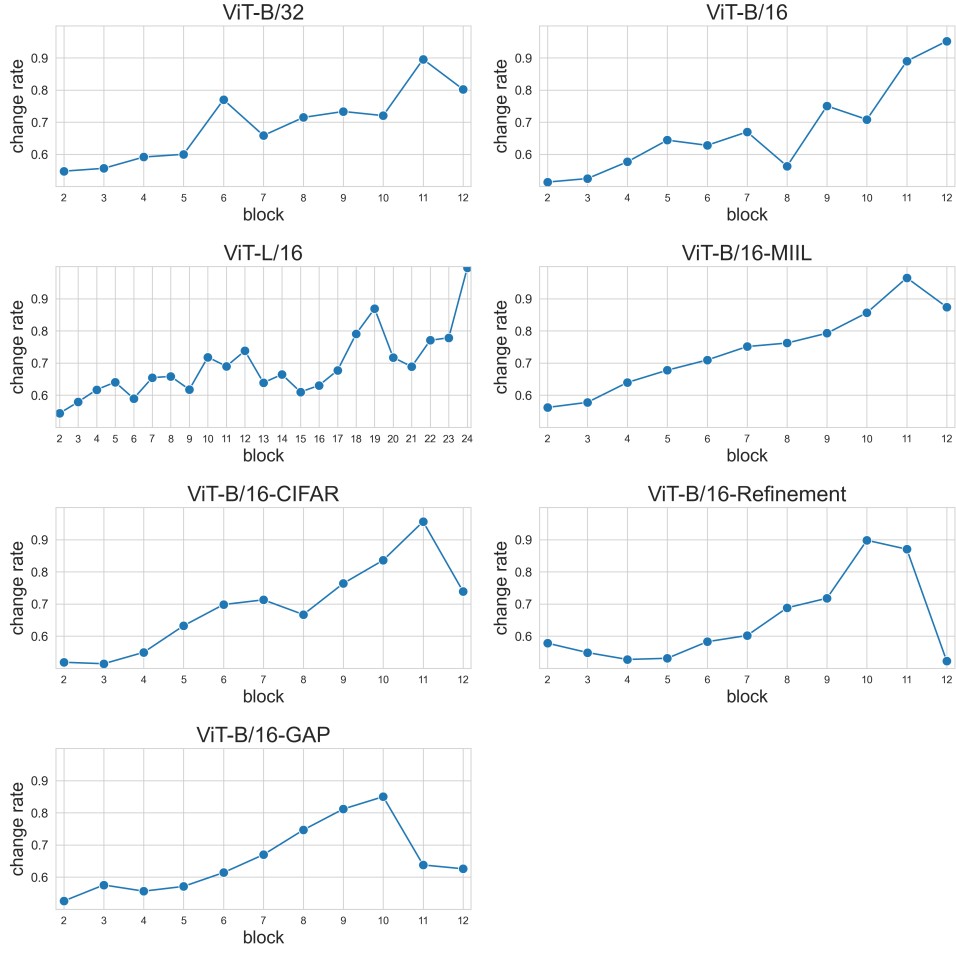

Figure 2: Percentage of image tokens that increment the logits of the correct class.

## 2.4 Class-labeled and context-labeled identifiability evolution

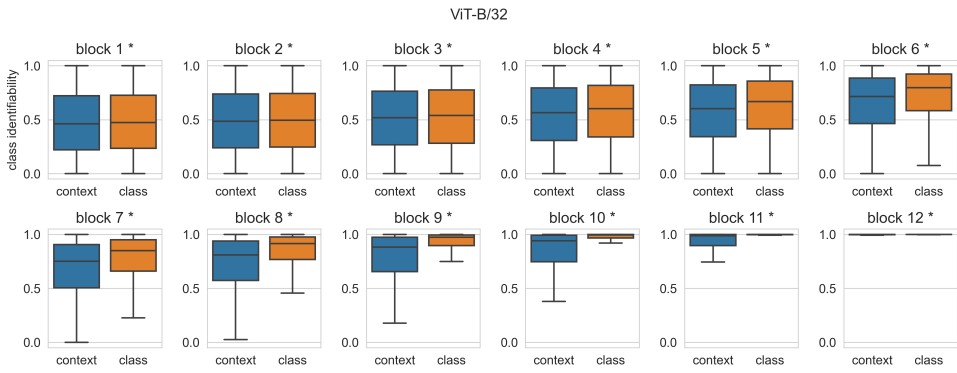

Figure 3: Class identifiability evolution of class and context-labeled image tokens in ViT-B/32. Asterisks indicate a significant difference between both types of tokens.

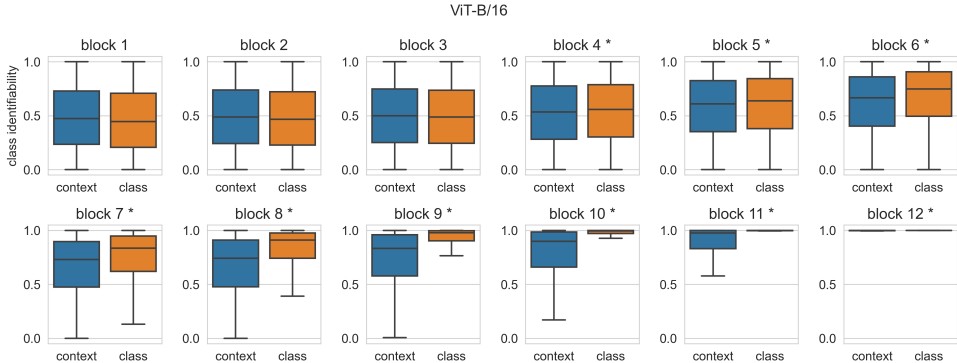

Figure 4: Class identifiability evolution of class and context-labeled image tokens in ViT-B/16. Asterisks indicate a significant difference between both types of tokens.

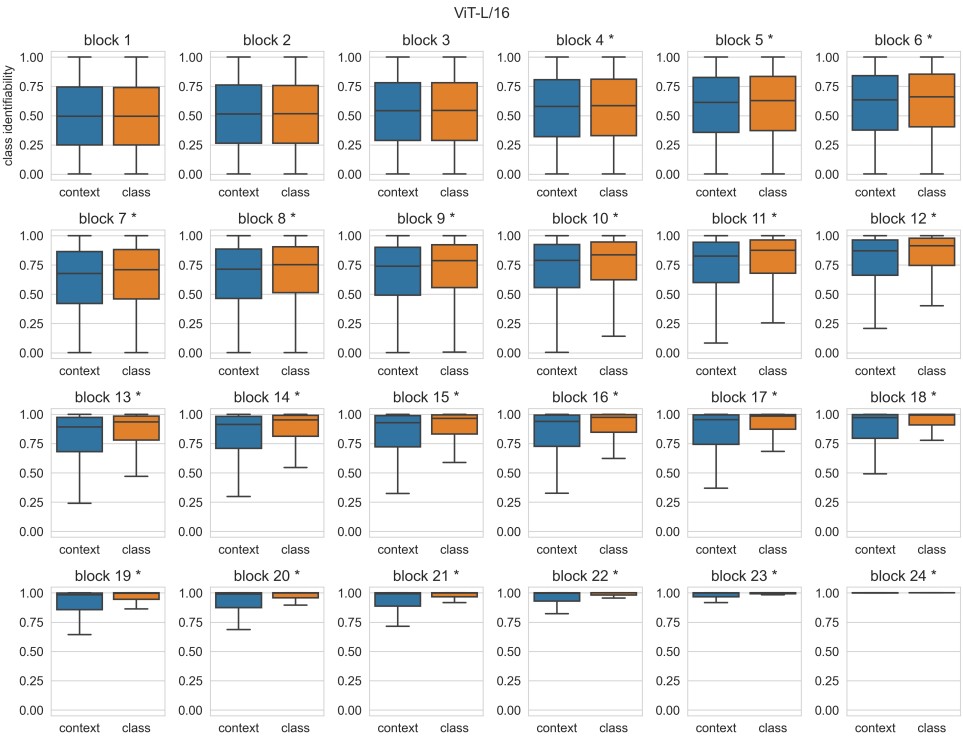

Figure 5: Class identifiability evolution of class and context-labeled image tokens in ViT-L/16. Asterisks indicate a significant difference between both types of tokens.

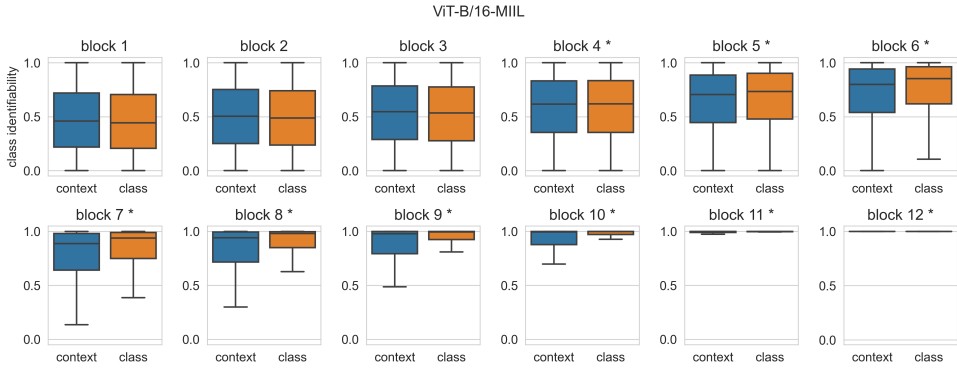

Figure 6: Class identifiability evolution of class and context-labeled image tokens in ViT-B/16-MIIL. Asterisks indicate a significant difference between both types of tokens.

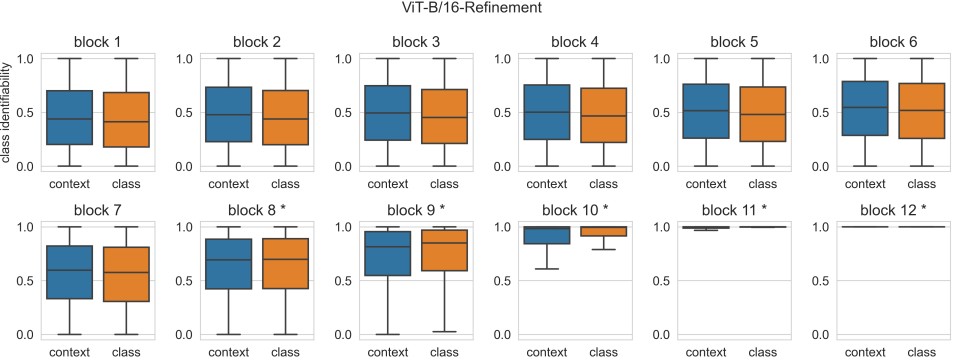

Figure 7: Class identifiability evolution of class and context-labeled image tokens in ViT-B/16-Refinement. Asterisks indicate a significant difference between both types of tokens.

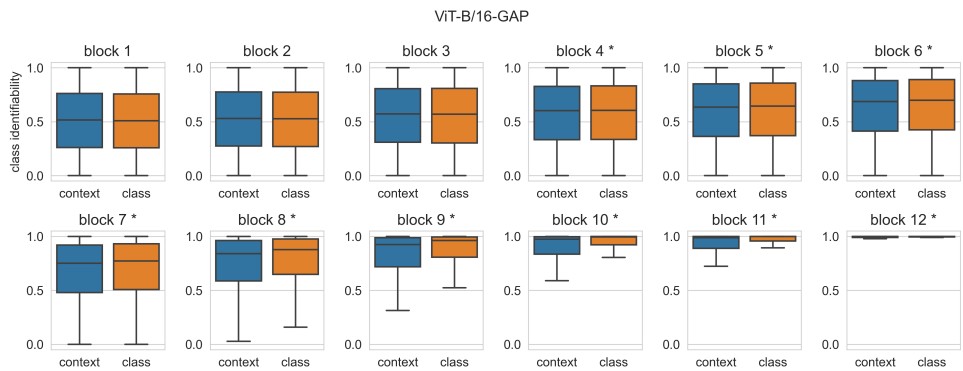

Figure 8: Class identifiability evolution of class and context-labeled image tokens in ViT-B/16-GAP. Asterisks indicate a significant difference between both types of tokens.

# 3 Mechanistic interpretability applications

## 3.1 Building of class representations

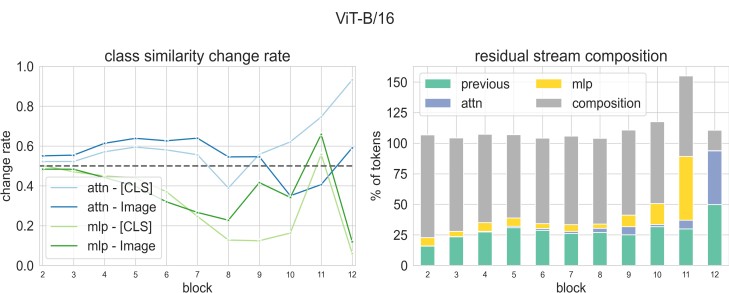

Figure 9: Building of class representations in ViT-B/16.

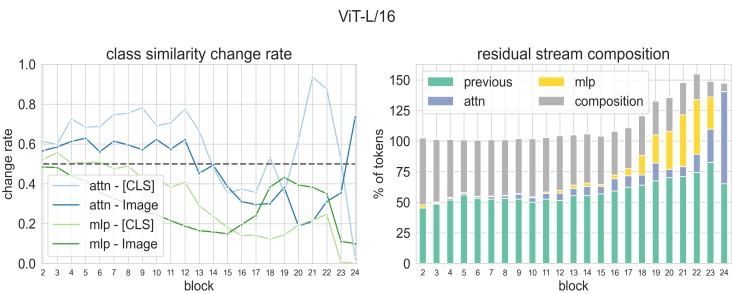

Figure 10: Building of class representations in ViT-L/16.

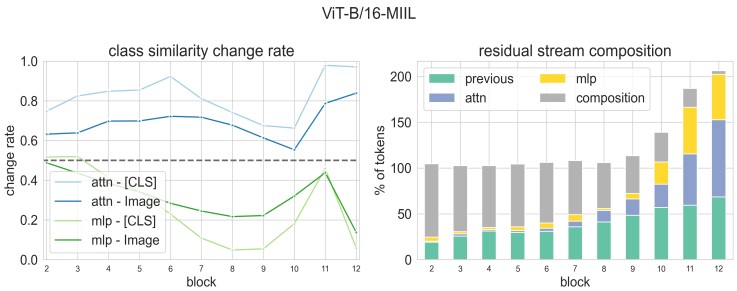

Figure 11: Building of class representations in ViT-B/16-MIIL.

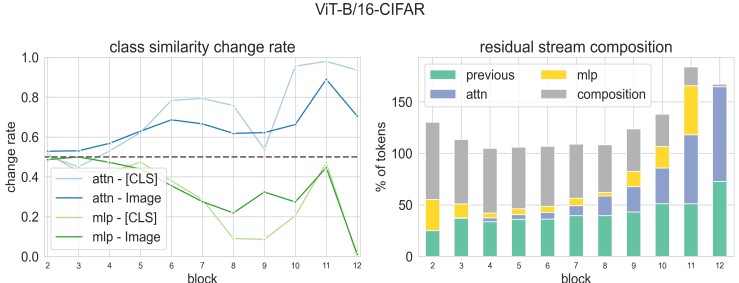

Figure 12: Building of class representations in ViT-B/16-CIFAR.

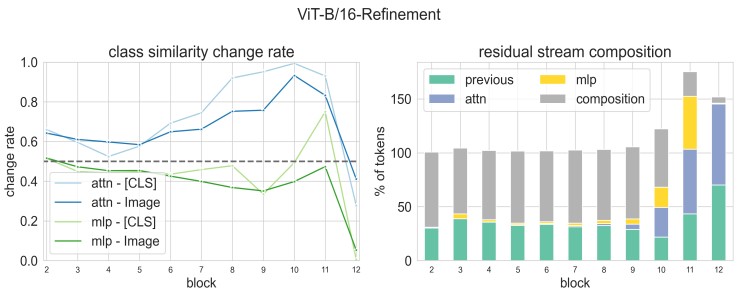

Figure 13: Building of class representations in ViT-B/16-Refinement.

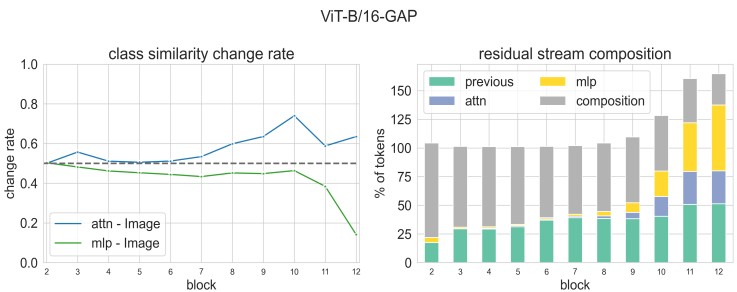

Figure 14: Building of class representations in ViT-B/16-GAP.

**Difference in class similarity change rate between class and context-labeled tokens.** We additionally conducted an analysis comparing the class similarity change rate of class- and context-labeled tokens in self-attention layers. We found that the increment is significantly higher for class-labeled tokens in the early and middle blocks, indicating that class-labeled tokens form categorical representations via attention mechanisms earlier than context tokens.

## 3.2 Categorical updates

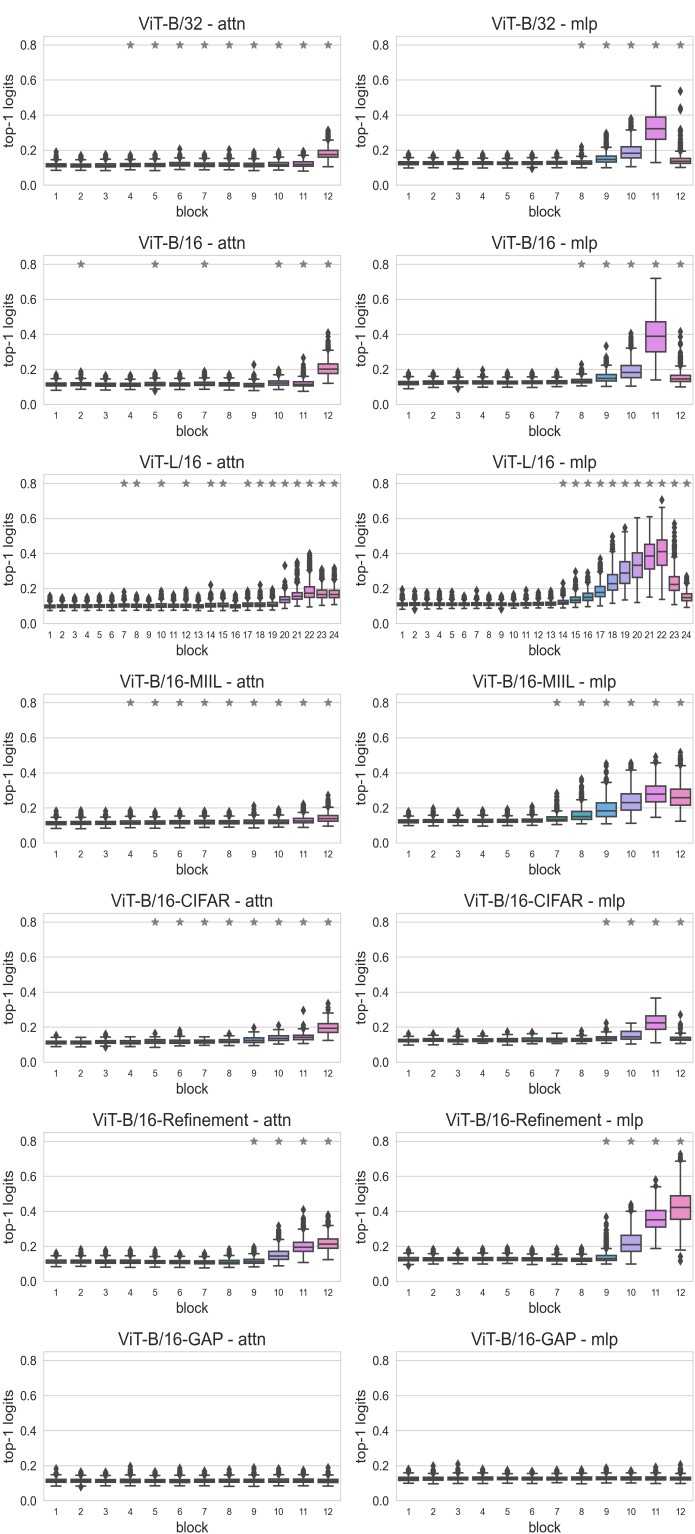

Figure 15: Class-value agreement scores. Asterisks indicate blocks where the scores were higher than those of a model initialized with random weights.

### 3.2.1 Human-interpretable value vectors

A manual inspection of the MLP layers containing high class-value agreement scores revealed that value vectors may promote and cluster semantically similar and human-interpretable concepts.

Table 2: Examples of most-activating classes for value vectors in MLP layer 11 of ViT-B/32.

| Memory index | Classes |
|---|---|
| $\mathbf{v}_4$ | racket, tennis ball, ping-pong ball, volleyball, croquet ball |
| $\mathbf{v}_8$ | arctic fox, brown bear, wombat, american black bear, wild boar |
| $\mathbf{v}_{10}$ | agama, whiptail, alligator, green lizard, american chamaleon |
| $\mathbf{v}_{32}$ | groenendael, scotch terrier, afghan hound, flat-coated retriever, newfoundland dog |
| $\mathbf{v}_{33}$ | strawberry, pineapple, tray, banana |
| $\mathbf{v}_{35}$ | rifle, revolver, assault rifle, scabbard, holster |
| $\mathbf{v}_{40}$ | wolf spider, garden spider, barn spider, harvestman, spider web |

Table 3: Examples of most-activating classes for value vectors in MLP layer 11 of ViT-B/16.

| Memory index | Classes |
|---|---|
| $\mathbf{v}_2$ | boathouse, paddle, water buffalo, gondola |
| $\mathbf{v}_7$ | microphone, radio, electric guitar, loudspeaker |
| $\mathbf{v}_{14}$ | screwdriver, carpenter's kit, plane, power drill, shovel |
| $\mathbf{v}_{18}$ | collie, border collie, kelpie, kuvasz, eskimo dog |
| $\mathbf{v}_{21}$ | guinea pig, beaver, hare, hamster, catamaran |
| $\mathbf{v}_{24}$ | rifle, assault rifle, revolver, bow, holster |
| $\mathbf{v}_{26}$ | drum, maraca, bell, steel drum, drumstick |

## 3.3 Key-value memory pairs at inference time

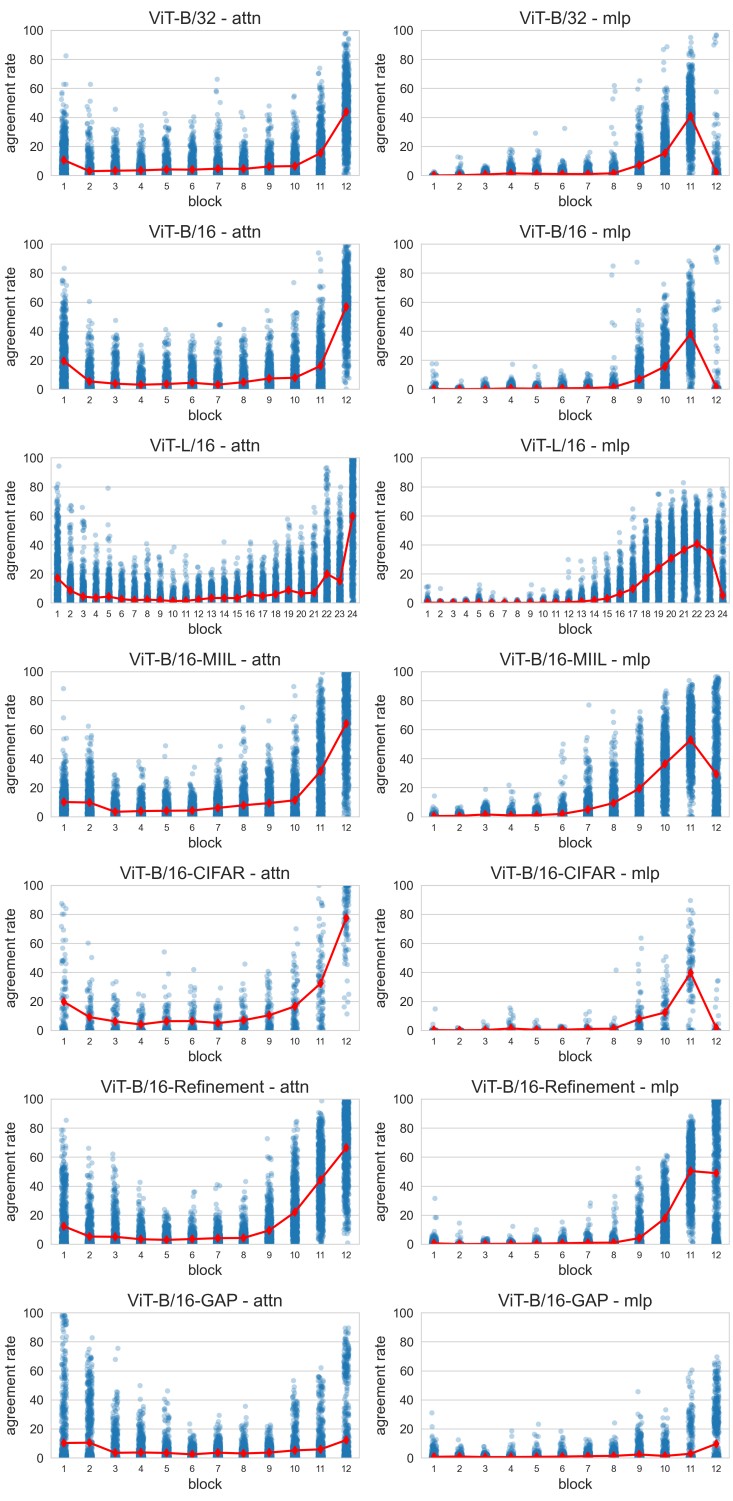

Figure 16: Key-value agreement rates.

### 3.3.1 Agreement rate influence in accuracy

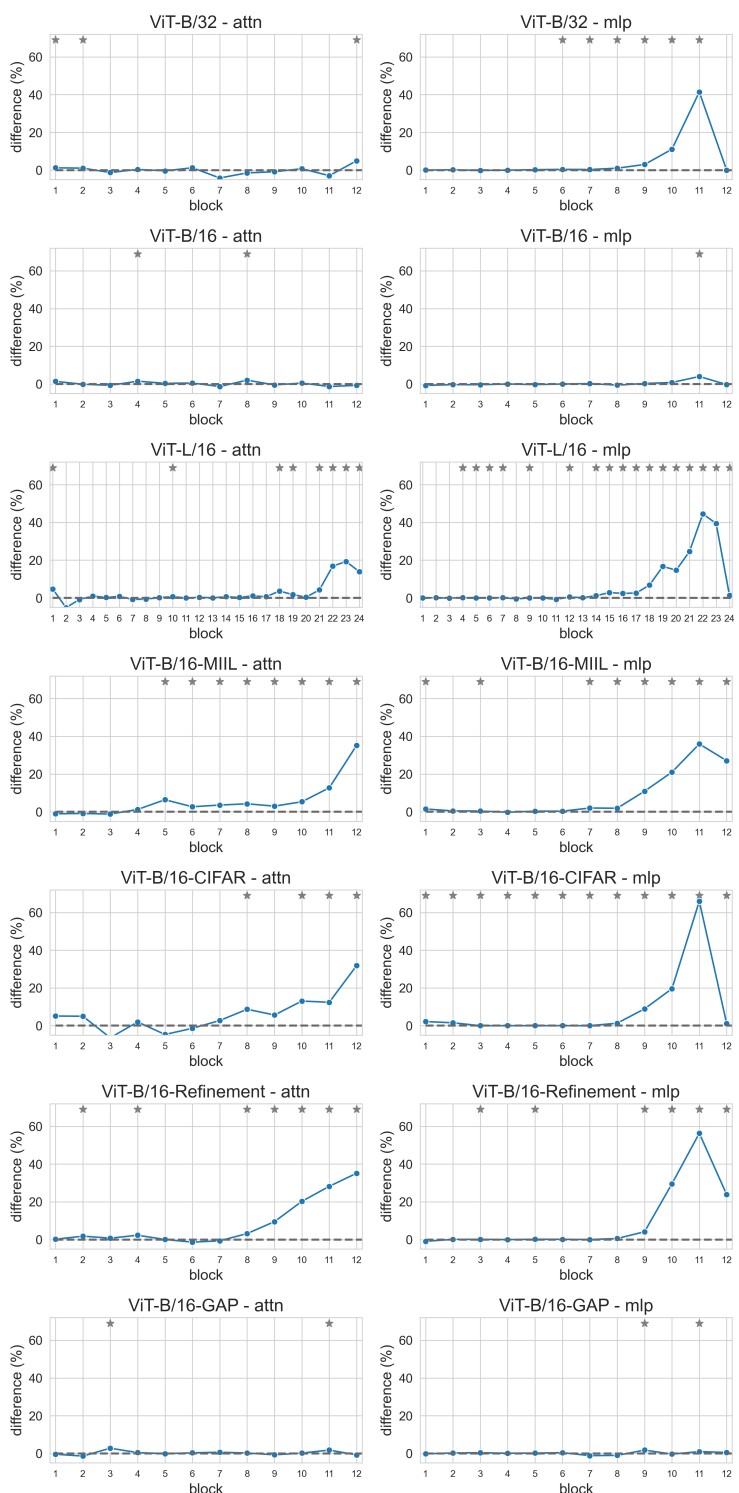

Figure 17: Agreement rate difference between correctly classified vs. misclassified samples. Asterisks indicate the blocks where the difference was statistically significant.

### 3.3.2 Compositionality of key-value memory pair systems

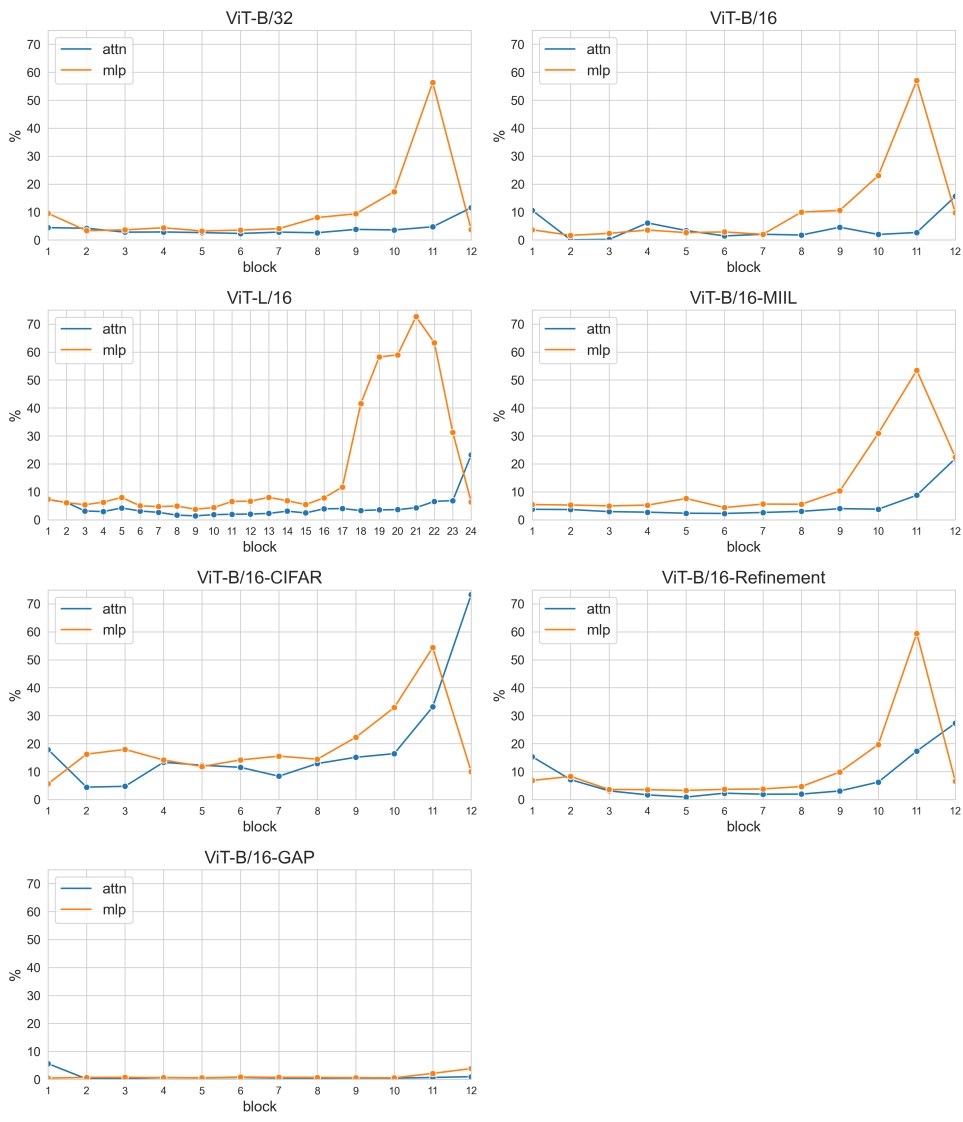

Figure 18: Percentage of instances where the layer's final predictions match any of the top-5 predictions of the most activated memories.

# 4 Explainability application

## 4.1 Additional feature importance visualizations

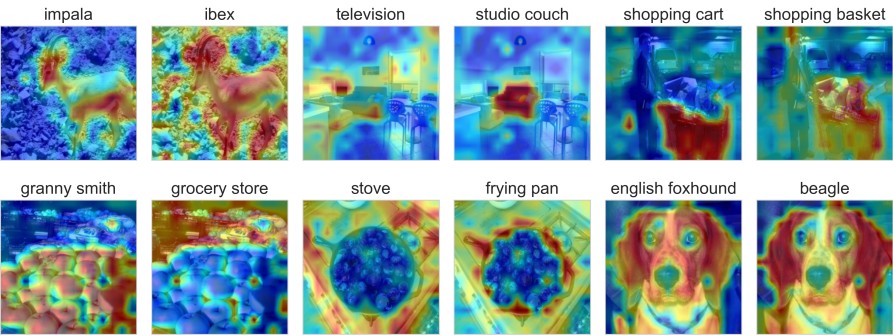

Figure 19: Examples of global feature importance visualization for ViT-B/16.

## 4.2 Perturbation studies of explainability framework

Table 4: *Results of negative and positive perturbation studies.*

|  | Negative | | Positive | |
| --- | --- | --- | --- | --- |
|  | Chefer et al. | Ours | Chefer at al. | Ours |
| AUC of accuracy scores | 84.54 % | 83.63% | 39.29% | 41.28 % |

# 5 Comparison with linear probing studies

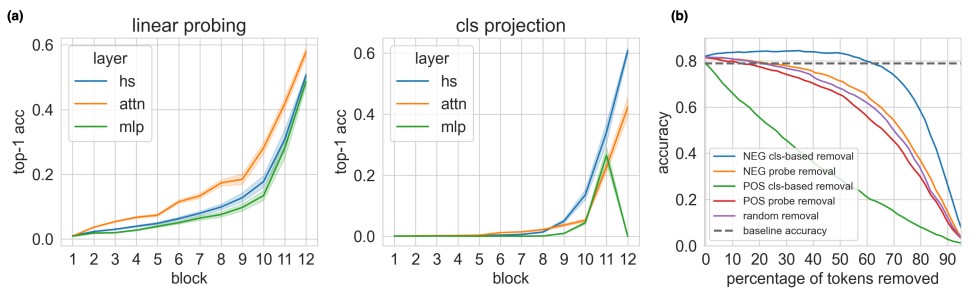

Figure 20: *Comparison with linear probing studies in ViT-B/32.* **(a)** Top-1 accuracy in the image classification task of both methods; **(b)** Perturbation Experiments. For the negative perturbation experiments (NEG), higher AUC is better, while in the positive perturbation experiments (POS) a lower AUC is better.