# OpenReview forum: "Analyzing Vision Transformers for Image Classification in Class Embedding Space"
_NeurIPS.cc/2023/Conference — NeurIPS 2023 poster_

### Official Review · Reviewer_vRFW · 2023-07-02

**Soundness:** 1 poor
**Presentation:** 3 good
**Contribution:** 2 fair
**Rating:** 3
**Confidence:** 4

**Summary:**

The authors propose a method that explores properties of vision transformer (ViT) features. In particular, tokens of patches at various levels of the sequence of transformer blocks are projected to class-space in the case of models pre-trained for image classification. Building of this, authors provide various insights into representations spaces of ViTs using quantitative experiments.

**Strengths:**

1) The paper explores an interesting and useful direction on better interpretability of ViTs
2) Methods proposed can be useful to community

**Weaknesses:**

1) Novel contribution over key related work [10,12] and additional insights in the form of quantitative experimental results seem insufficient to provide any strong information on ViTs
2) Projecting intermediate features to class space with a weight matrix from the last layer is not  directly meaningful - maybe explain the reasoning / assumption why this would provide any useful information? The differences in results across layers could be due to misalignment (between feature and projection weights) and not necessarily lack of information.
3) How does explicit aligning of intermediate representations to class space (e.g. see [1]) affect findings? Maybe linear projection of those features to better verify what information is contained within them?
4) Results and discussion in section 5 are not explained well

[1] Naseer, Muzammal et al. “On Improving Adversarial Transferability of Vision Transformers.” ArXiv abs/2106.04169 (2021): n. pag.

**Questions:**

The statement “we can project the internal representations of ViT into the class embedding space to probe their categorical representations” is not well supported by explanation or results (see weaknesses above for more). This is assumption is central for almost all insights provided by paper. In fact, this leads to multiple missing links in various statements made in paper (and explanations given to results).

**Limitations:**

No clear discussion of limitations or possible impacts.

---

> ### Author Rebuttal · Authors · 2023-08-09
>
> Thank you for your feedback which has helped 1) clarify in our manuscript the rationale, relevance, and novelty of our work, and 2) improve the generalizability and clarity of our findings. We address in detail each concern stated in the revision below.
>
> _On the rationale behind our method_
>
> We argue that projecting intermediate representations to the class-embedding space allows us to investigate how the hidden states increasingly represent the class prototype learned by the model. This assumes that the class prototype is encoded in the embedding weights of the class-projection matrix (an assumption that has been made before, see for example [1]). The new experiments comparing our framework with linear probing studies (see general response) give evidence that our method can better detect the features that lead to a categorically meaningful representation. Note that we do not use our framework to claim that early layers lack information that, after linear and non-linear transformations, will be relevant for making categorical decisions. In fact, we show that from very early layers of the model (including layer 1 for some ViT variants) there is significant alignment to the categorical representation as compared to a random model. Instead, our framework shows how projection to the class embedding of intermediate features allows us to investigate the factors (i.e. token position) and mechanism (i.e. via key-value memory pair mechanisms) that increase the alignment with the class prototype.
>
> We have clarified these issues in the new version of our manuscript.
>
> _On the novel contributions of our work_
>
> Regarding the novelty of our contributions over [10,12], the key additional advances of our work are the following:
> - We introduce a generalizable interpretability framework that can be used to efficiently investigate the categorical building processes of ViTs. We strengthen the generalizability and advantages of our framework by using the new experiments reported in the general response to show that: (1) We can apply our method to analyze the representations of ViTs that have learned a class-embedding matrix during training; (2) Our framework can distinguish the effects of different training variants (e.g. datasets, architectural depth or constraints) in the build-up of class representations; (3) Our method can characterize how categorical information emerges in the image tokens more efficiently and accurately than the commonly used linear probe method (see next section for additional details).
> - We use our framework to investigate how a specific mechanism not mentioned in [10,12], namely key-value memory pair systems, is used by different layers of ViTs to add semantically meaningful information to the residual stream and build categorical representations. Previous work in NLP showed that these mechanisms can be exploited for model editing, efficiency improvements at inference time, or performance improvements, and thus we consider that our study opens future avenues of research. Moreover, our new experiments (see general response) showing that these mechanisms are present across ViT variants also give evidence of the broad impact of understanding key-value memory pair systems.
>
> We clarify and reinforce the importance of these findings in the “Introduction” section of the new version of our manuscript.
>
> _On the comparison with linear probing methods_
>
> In the general response, we provide a comparison of our framework with that of linear probing studies. In these experiments, we specifically show how quantifying the “class-relevant” information contained in the inner representations of ViTs with linear probes does not necessarily enable insights into the features and mechanisms that the model uses for making categorical decisions. This is in contrast with our method which enables such insights.
>
> _On the analysis of ViTs with explicit intermediate alignment to class representations_
>
> Thank you for suggesting applying our framework to ViTs trained by Naseer et al., as it has led to meaningful insights and has strengthened the generalizability of our work.
> As shown in Table 1 and Fig. 1 of the rebuttal PDF, results suggest that aligning the intermediate representations of the tokens to the class embedding space leads to 1) increased class-identifiability scores in [CLS] tokens across all layers; 2) more class-identifiable image tokens in the last layer; 3) less class-identifiable tokens in middle layers. Moreover, as shown in Fig. 2 of the rebuttal PDF, these ViTs also make use of key-value memory pair systems and, unlike other networks, extend this use to block 12. The difference might be due to the use of the image tokens in MLP layer 12 to predict the correct category in the class-refinement module, compared to common ViT variants where only the [CLS] token in MLP layer 12 is used for prediction.
>
> _On the clarity of Section 5_
>
> We have now improved the discussion of the results in Section 5, and we:
> - Explain some of our results in light of previous work, and provide more interpretation of the patterns that we observe.
> - Rephrase some of our statements to improve clarity.
> - Improve the captions of Table 1.
> If there are other concrete concerns with the clarity of this section other than those mentioned, we will be glad to further work on them.
>
> _On the discussion of limitations_
>
> Note that we do mention the limitations of our work and its impact in the original manuscript; specifically in the original conclusion section. If a specific aspect that the reviewer considers important and was overlooked in the description of our limitations, we are glad to include it in the new version of our manuscript.
>
> We have benefited from your comments and suggestions, and believe we have addressed them favorably. If so, please consider raising the reviewer rating.
>
> [1] Hill, F. “Why transformers are obviously good models of language.” (2023).

---

> > ### Comment · Reviewer_vRFW · 2023-08-15
> > **Response to reviewer**
> >
> > I thank the authors for the rebuttal.
> >
> > However, my two key concerns, weaknesses 1 and 2, remain unresolved. I keep my rating as it is.
> >
> > On weakness 2, *"This assumes that the class prototype is encoded in the embedding weights of the class-projection matrix (an assumption that has been made before, see for example [1])."*, this assumption still is vague with references to non-peer reviewed works attempting to support it.

---

> > > ### Author Response · Authors · 2023-08-16
> > >
> > > Thank you for your feedback. We would appreciate making the concerns concrete (please see below).
> > >
> > > Regarding weakness 2, we want to clarify that our assumption (actually, an interpretation) is both theoretically and empirically supported.
> > >
> > > It is supported theoretically by the fact that the final classification layer is only reading information from the class-embedding weight matrix, which thus has to encode the patterns that allow the network to predict each class. We assume these patterns reflect the class prototypes.
> > >
> > > This interpretation is trivially true courtesy of the image classification mechanism of the transformer architecture. The decision is a function of the dot product between the [CLS] token in the hidden state of the last layer and each row (“class prototype”) in the class embedding matrix. The probability of the image belonging to a category is proportional to this dot product. Our interpretation in terms of “class prototype” follows directly from the definition of dot product, since the latter will reflect how close the [CLS] token is to a class row.
> > >
> > > Our interpretation is further supported empirically in our experiments given that the alignment of each token to the class-embedding projection is predictive of the relevance that this token has in the categorical decision, as demonstrated in our perturbation studies.
> > >
> > > More broadly, it would be useful to better understand why the reviewer’s concerns remain unresolved. In responding to weaknesses 1 and 2, in our rebuttal and revised manuscript (1) we clarified the novelty of our work by pointing out two broad contributions that are absent in the literature, and (2) we explicated the rationale of our method, in particular, why our method is informative for our research goals and why linear probes are not appropriate for such purposes (please see rebuttal for details).
> > >
> > > To improve the quality of our work, we would appreciate it if concerns with the statements made in our rebuttal of weaknesses 1 and 2 could be made concrete.

---

> > > > ### Comment · Reviewer_vRFW · 2023-08-18
> > > > **Unsupported / Weak Claims**
> > > >
> > > > **Weakness 2**:
> > > > `Projecting intermediate features to class space with a weight matrix from the last layer is not directly meaningful`
> > > > * Consider the output space of the network's last transformer block. Within this space, the weight matrix of last linear layer (operating on the outputs of that transformer block to project to class distribution space) represents class prototype embeddings. This is what the cited supporting work claims (and can be assumed true, for what matters here).
> > > > * However, in a different space, this assumption would not hold. For example, taking the last linear layer weight matrix of one network, and probing the outputs of different network (with common dimension outputs) will not give the same results. In the same way, although intermediate layers have a common dimension, these vectors lie in a different space.
> > > > * Claiming that projecting these vectors to class distribution space with a mismatched weight matrix results in low performance is therefore unfair. While the original vectors may not contain class information, it could also be that the weight matrix used is mismatched with the space in which these vectors lie.
> > > > * In the main rebuttal, when dismissing linear probing, the authors discuss possibility of spurious background correlations affecting linear probing, but there is no reason why the learned weight matrix of last layer would not be similar. In fact, even methods like CLIP have been shown to contain such spurious correlations. This actually raises a secondary issue of how correct (absence of spurious correlations) the final layer weight matrix is, to serve as good class prototypes.  More importantly, the conflicting results of linear probing (whose theoretical weaknesses are all shared by the last-layer weights method), raise further concerns.
> > > >
> > > >
> > > > **Weakness 1**:
> > > > Given the weakness in above assumption, much of the proposed insights into network weights are rendered unsubstantiated. This leaves with minimal additional contributions over the existing works such as [10, 12].
> > > >
> > > >
> > > > **Final Response:**
> > > > I am quite aware about contemporary work in this topic and consider myself to have a decent understanding of those works. However, in light of highly positive comments by other reviewers, and given the chance that I may have misunderstood specifics of this paper, I will change my vote to Reject (as opposed to Strong Reject). I remain at Reject because the concerns discussed above have not been sufficiently addressed or discussed in the main paper or rebuttal. Further, most results figures and tables in the paper (and even rebuttal) still appear unclear (they are not described adequately and they are not self-explanatory either).

---

> > > > > ### Author Response · Authors · 2023-08-21
> > > > >
> > > > > Thank you for your clarifications. We think there are some misunderstandings about what our paper does and claims. We discuss these below.
> > > > >
> > > > > > However, in a different space, this assumption would not hold. For example, taking the last linear layer weight matrix of one network, and probing the outputs of different network (with common dimension outputs) will not give the same results. In the same way, although intermediate layers have a common dimension, these vectors lie in a different space.
> > > > >
> > > > > This implies that in our framework we need to assume that the intermediate layers lie in the same space as the class-embedding space, in order to conduct later analysis. This is not the case for our paper.
> > > > >
> > > > > Our framework does not need to assume alignment: the method quantifies this alignment. Our results show that there is an increasing alignment to the class-embedding space across the hierarchy of the network. We show this progressive alignment is meaningful through perturbation studies.
> > > > >
> > > > > The focus of our paper is to understand the mechanisms in the network (namely key-value memory pair systems) which enable this increasing alignment.
> > > > >
> > > > > > Claiming that projecting these vectors to class distribution space with a mismatched weight matrix results in low performance is therefore unfair. While the original vectors may not contain class information, it could also be that the weight matrix used is mismatched with the space in which these vectors lie.
> > > > >
> > > > > We do not make such a claim in our paper. What we claim is that we found that the degree of the activation of the key-value memory pair system in MLP layer 11 (which adds more class-aligned representations to the residual stream of ViT) correlates with performance.
> > > > >
> > > > > > In the main rebuttal, when dismissing linear probing, the authors discuss possibility of spurious background correlations affecting linear probing, but there is no reason why the learned weight matrix of last layer would not be similar. In fact, even methods like CLIP have been shown to contain such spurious correlations. This actually raises a secondary issue of how correct (absence of spurious correlations) the final layer weight matrix is, to serve as good class prototypes. More importantly, the conflicting results of linear probing (whose theoretical weaknesses are all shared by the last-layer weights method), raise further concerns.
> > > > >
> > > > > We indeed agree that is likely that the class-embedding matrix also encodes spurious correlations. However, these are the spurious correlations the network is learning to associate with the class in its final decision. In other words, these are the biases the network is learning with each class prototype.
> > > > >
> > > > > Identifying these spurious associations is important since they point to which features are relevant for improving the semantic robustness of ViT. And these features are not the ones identified with vanilla linear probing, as shown in the general rebuttal.
> > > > >
> > > > > We disagree that our results being different from linear probing is problematic since they are fundamentally tackling different research questions, as explained in the general rebuttal.
> > > > >
> > > > > > Weakness 1: Given the weakness in above assumption, much of the proposed insights into network weights are rendered unsubstantiated. This leaves with minimal additional contributions over the existing works such as [10, 12].
> > > > >
> > > > > We respectfully disagree with the reviewer on this point, for the reasons mentioned in rebuttal 1 and in response to other reviewers.
> > > > >
> > > > > > Further, most results figures and tables in the paper (and even rebuttal) still appear unclear.
> > > > >
> > > > > As stated in the first rebuttal, we will change the captions of Table 1 to contain more details about their content. We would also add better captions to the figures and work on their visibility.

---

### Official Review · Reviewer_GYsr · 2023-07-05

**Soundness:** 3 good
**Presentation:** 3 good
**Contribution:** 3 good
**Rating:** 6
**Confidence:** 4

**Summary:**

The authors propose to reverse-engineer pre-trained ViTs for image classification task in order to investigate how the internal representations at different levels are projected onto the class embedding space and reveal how the models construct representations for predictions. It provides insights into the distinct contributions of self-attention and MLP layers in ViTs to the categorical composition. The proposed method can further identify important image regions for class detection as a valuable toll for achieving mechanistic interpretability and explainability in ViTs.

**Strengths:**

1. This paper presents a pioneering approach to reverse-engineering pre-trained ViTs for image classification tasks, offering new insights into how ViTs construct representations for their predictions. While the concept of reverse-engineering is inspired by NLP research, this is the first work to apply it specifically to ViTs in computer vision tasks.
2. The authors introduce a framework that enhances mechanistic interpretability and explainability in ViTs, enabling the identification of the most relevant image regions for detecting a specific class of interest.
3. The paper emphasizes the distinct roles of self-attention and MLP layers in this process, illustrating how they contribute differently to categorical updates by utilizing compositional key-value memory pair mechanisms.
4. To evaluate their findings, the authors employ several metrics including the class-value agreement score, key-value agreement rate, class similarity change rate, and match between the top-1 prediction.
5. The paper is highly accessible with clear logical flow. The tables and figures are presented in a manner that is easy to read and understand, contributing to the overall clarity of the research.

**Weaknesses:**

1. The findings might differ when using this method to reverse engineer transformers that are larger, have been trained with different datasets, or contain architectural modifications. For example, the paper focuses on vanilla ViTs trained on ImageNet, and it is unclear how well the proposed framework generalizes to other types of ViTs or other datasets
2. The paper does not provide a comprehensive comparison with other methods for interpretability or explainability of ViTs, although it does mention some related work in this area.
3. Certain concepts require additional investigation. The reasons behind the distinct performance of block 11 compared to other blocks, as well as the rationale for summing the gradients across the blocks, remain unclear and warrant further study.

**Questions:**

1. Why there are performance differences betwee block 11 and other blocks?
2. Why sum the gradients over the blocks?
3. What does it mean in line 307-308?

---

> ### Author Rebuttal · Authors · 2023-08-09
>
> Thank you for the careful reading and useful feedback. We address the comments and questions below.
>
> _Weakness 1_
>
> Thank you for the suggestion. We have now demonstrated the generalizability of our method to all cases mentioned by the reviewer (see general response) and expanded our analyses to other variants of ViTs, including larger models, trained on different datasets, or containing architectural modifications (e.g. use of GAP). The new experiments demonstrate that we can apply our method to investigate a wide range of ViTs.
>
> _Weakness 2_
>
> Thank you for suggesting comparisons with other mechanistic interpretability and explainability methods.
>
> Regarding mechanistic interpretability, we have added a comparison with linear probing approaches (see general response), which are widely used to study if categorical information is present in the hidden states of DNNs. Our new experiments show that the insights we can obtain from linear probes are very different from those of our method, and less informative for the purposes here.
>
> Regarding explainability, our approach is complementary to existing methods and provides different kinds of insights. Specifically, SOTA methods that generate relevancy maps for a classification example compute the gradients of the class logits at the final layer with respect to the input and/or aggregate the information propagated across layers up to the class-embedding projection. Instead, our method is able to visualize the categorical information contained in the image tokens independently for each block. These block-specific visualizations allow us to (1) better understand how categorical information is hierarchically built, and (2) characterize the importance of each block in building the class representations. We clarify this conceptual difference between methods in the new version of our paper and provide some examples of layer-specific visualizations.
>
> Regarding question 2, we originally suggested aggregating the gradients over blocks to show that our framework can also provide a global relevancy map that is semantically meaningful and can be used for traditional explainability visualizations. The sum procedure also allows us to corroborate that we obtain a fair portrayal of the individual contribution of each block to the final categorical representation: if the aggregated visualization provides a coherent and useful relevancy map (measured with perturbation studies), there is evidence that the interpretation of the individual blocks is meaningful. We have explained why we carried out this sum explicitly in the new version of our paper.
>
> In addition, we compare the quality of our global relevancy map to that of an established explainability method (reference [3] in our paper) to further prove the accuracy and quality of our proposal. Using ViT-B/32, for both methods we (1) quantified the importance of each token, (2) gradually removed the tokens with the least to most importance scores (negative perturbation test), (3) gradually removed the tokens with the most to least importance (positive perturbation test), (4) measured the accuracy of the model with each removal, (5) computed the Area Under the Curve of the final accuracies. Our results show that our methods yield similar results to those of [3] (Table 2 in rebuttal PDF), highlighting the adequacy of our approach.
>
> _Q1_
>
> We found that in VIT-B/32, MLP 11 promotes the strongest categorical updates via key-value memory pair mechanisms (section 6.2). This could explain why if the activation of the key-value memory pair system in MLP 11 is high (as measured by the key-value agreement rate), a better classification accuracy will be obtained as a result of a stronger categorical update.
> Regarding why the strongest categorical updates are encoded in MLP 11, we hypothesize it is due to this being the last layer from which the [CLS] token can extract information (through the self-attention layer in block 12) to make a categorical decision. In turn, MLP layers promoting stronger categorical updates at the very final layers could be useful because the depth allows the model to develop more complex and semantically meaningful keys (as shown in NLP [1]).
> Our new experiments provide additional evidence in favor of these hypotheses: 1) Models with deeper architectures present stronger categorical updates in the layers before the final one, not in layer 11 (Fig. 2 in reb. PDF); 2) When training explicitly aligns intermediate representations to the class embedding space in every layer, and thus the network is able to use the image token representations in MLP 12 to make categorical decisions, we found that the strength of the categorical updates in MLP 12 is even higher to those of MLP 11 (see Refinement ViT in Fig.3 of PDF).
> In addition, we also note that in the self-attention 12 the strength of the categorical updates and the activation of key-value memory pair systems is higher than those of earlier blocks. This is further evidence that ViTs exploit key-value memory pair systems for promoting categories at the very late stages of the network so that the key representations can encode greater semantic complexity.
>
> We have added this interpretation of the results to our paper.
>
> _Q2_
>
> See discussion of weakness 2.
>
> _Q3_
>
> If the question refers to our compositionality results, another way of stating these findings would be: in at least 80% of the cases, the final predictions of these layers do not exactly match the categories promoted by the most activated keys. Instead, results suggest that in the majority of cases, the final prediction is a composition of the categorical representations promoted by more than one key. The strength of this compositionality varied across blocks and layers.
>
> We thank you for the suggestions again, which we consider to have been addressed successfully. If so, please consider raising the reviewer rating.
>
> [1] Geva et al. "Transformer feed-forward layers are key-value memories." (2020).

---

> > ### Comment · Reviewer_GYsr · 2023-08-20
> > **Thank you for detailed clarifications.**
> >
> > Thank you for detailed clarifications. Some of my questions have been addressed. I am inclined to raise my score.

---

### Official Review · Reviewer_9G5D · 2023-07-06

**Soundness:** 4 excellent
**Presentation:** 4 excellent
**Contribution:** 3 good
**Rating:** 7
**Confidence:** 5

**Summary:**

This work analyzes how Vision Transformers work by analyzing the representations of individual tokens (image patch representations) and how they evolve while passing through the layers of the network. The authors also show how to use their methods to devlop an interpretability method.

**Strengths:**

Originality: The work applies experiments originaly proposed in the NLP space to analyse Transformers to the Vision Transformer. I am not aware of anyone having done these types of experiments for ViT before, and the insights gained this way are interesting.

Quality: The experiments seem solid, and while I have doubts about a small part of them (see weaknesses) they are overall well done. They are well thought out and test simple hypotheses. The value of these experiments has already been validated in the NLP space.

Clarity: I found the experimental design was clear and the exposition easy to follow.

Significance: I think this work contributes to our understanding of how Vision Transformers work. In themselves they don't offer completely new insights, but confirm existing knwoledge/intuition/theories and add empirical evidence that aid our understanding. I think the work is easily understandable and accessible, and gives insights that help understand the inner workings of one of the currently most-used models. I think the contribution is valuable.

**Weaknesses:**

* The paper investigates the original ViT architecture as proposed in the 2021 ICLR paper. It completely ignores that we know now that ViT does not need a CLS token (I think this was first proposed in "Scaling Vision Transformers" by the original authors of ViT in shortly after their first paper). This paper independently confirms this finding, but simply citing that paper would have been easier. Also, by analysing a ViT model that used a Global Average Pool (or MAP) to classify  would remove one additional cofounder. I'd appreciate if the authors could briefly mention/discuss this point in the next version of the paper (I feel like  part of the reason people keep using CLS-token ViT is that every other paper does it, so I'd encourage the authors to point out that this isn't needed any more).

* A potential relevant related work is "Understanding Robustness of Transformers for Image Classification", ICCV 2021, Bhojanapalli et al.,. The 2nd half of that paper also tries to understand how ViT work by ablating parts of the modle (e.g. by removing individual Self-Attn. and MLP layers), which feels similar to the pertubation studies presented in Table 1.

**Questions:**

No questions come to mind.

**Limitations:**

The authors do not discuss limitations of their work, but given the nature of the paper (empirical exploration instead of proposing a new method) this does not apply as much.

---

> ### Author Rebuttal · Authors · 2023-08-09
>
> We thank the reviewer for their positive feedback and suggestions for improving the discussion of our findings. Below we address each of these in detail.
>
> _Weakness 1_
>
> Thank you for pointing out relevant previous work of [1]. We have added a reference in section 5.1 of our manuscript, stating in line 195 that “These results are aligned to those of [1], where it is shown that ViTs trained without the [CLS] token can achieve similar performance to models that include it”.
> We also want to highlight that our experiments are not redundant but complementary to [1] in the following sense. Our results show that even in ViTs trained with the [CLS] token, the image tokens can achieve class decodable information without extracting information from the [CLS], which sheds light on the mechanisms behind the building of categorical representations.
>
> Moreover, as described in the general comment, we carried out additional experiments and used our method to probe ViTs trained with GAP. We found that the training with GAP changes the mechanisms by which the network creates categorical representations. Concretely, while introducing GAP does not decrease the class identifiability scores of image tokens across layers (see Fig. 1 of the rebuttal PDF), it decreases the class identifiability rate of the tokens in the last layer (i.e. the percentage of tokens that contain a class identifiability score of 1). In addition, as Figure 3 of the rebuttal PDF shows, GAP training decreases the reliance on key-value memory pair mechanisms for building categorical representations. Taken together, these findings corroborate the idea that the categorical representations on GAP-based ViTs emerge in a more distributed fashion than in [CLS]-based ViTs. We have added these findings and their discussion to the new version of our manuscript.
>
> _Weakness 2_
>
> Thank you for highlighting the relevant work by Bhojanapalli et al. (now cited)
>
> Our analyses are complementary to those of Bhojanapalli et al. who investigate the effects of removing self-attention and MLP layers in the performance of ViTs. In contrast, we analyze in detail how the categorical representations of ViTs are modified and built by these sub-modules, regardless of performance. We concretely investigate the use of key-value memory pair systems in these processes.
>
> In addition, we think our perturbation studies complement the findings on the robustness of ViTs in Bhojanapalli et al. We show how context-tokens are necessary for building a categorical representation in the class-labeled image tokens, which points to how much information unrelated to the class the model is including in the category representation of the embedding matrix.
>
> We have added a discussion of the work of Bhojanapalli et al. and the mentioned differences with our study to the “Related Work” section.
>
> Thanks again for the feedback. We believe we have addressed all your concerns. If so, please consider raising the reviewer rating.
>
> [1] Zhai et al. "Scaling vision transformers." (2022).

---

> > ### Comment · Reviewer_9G5D · 2023-08-15
> >
> > Thank you for the clarifications and the additional experiments. I stand by my original review that this work should be accepted to the conference.

---

### Official Review · Reviewer_mxXe · 2023-07-07

**Soundness:** 3 good
**Presentation:** 3 good
**Contribution:** 3 good
**Rating:** 6
**Confidence:** 3

**Summary:**

Inspired by recent advancements in NLP, this paper introduces a novel framework designed to reverse engineer vision transformers for the purpose of image classification tasks. The framework focuses on analyzing the internal dynamics of Vision Transformers (ViTs) within the class-embedding space, revealing the intricate process by which ViTs construct categorical representations using self-attention and MLP layers. The empirical findings gleaned from this investigation offer valuable insights into the inner workings of ViTs. Furthermore, the proposed framework can be utilized to identify the crucial components within an image that play a significant role in class detection.




**Strengths:**

1. The research presented in this paper enhances the current understanding of interpretability and explainability in ViTs. While previous studies have primarily examined the preservation of spatial representations in ViTs throughout the hierarchy, this work specifically investigates the construction of categorical information for the final prediction.

2. This research demonstrates the intriguing phenomenon of internal disruption in categorical representations caused by context and attention perturbations.

3. The Experiment Design in this study encompasses a comprehensive range.

**Weaknesses:**

1. It would be beneficial to include a thorough discussion in Section 3 regarding the distinctions between the proposed framework and similar work in NLP, which also involves projecting the internal representations of these models onto the output space. Including a comparison of the technical details between the two approaches would enhance the clarity of the paper.

2. The empirical results provided in the study primarily rely on the analysis of ViTs pre-trained on ImageNet. To strengthen the findings, it would be advantageous to incorporate results obtained from ViTs pre-trained on larger datasets. This expansion would offer a broader perspective and further validate the conclusions drawn in the research.

**Questions:**

See the weakness part.

**Limitations:**

In Section 8, the authors discuss the limitations of the adopted model's diversity and the underexplored application for performance improvement.

---

> ### Author Rebuttal · Authors · 2023-08-09
>
> Thank you for the positive remarks and suggestions for improving the clarity and generalizability of our work. Below we address these comments in detail.
>
> _Weakness 1_
>
> Thank you for the concrete suggestion on how to improve the description of our method. In our revised version of the paper, we have added a subsection thoroughly comparing our approach with that of NLP.
> Due to spatial constraints, we cannot reproduce the entire subsection here. As a summary of the added information in this section:
> - We describe how the input- and output-embedding matrices of both types of networks differ. Pre-trained LLMs include a vocabulary-projection matrix that serves both as the input and output embeddings of the model. In contrast, the nature of the input and output embedding matrices differ in ViTs: the input embedding matrix is a linear projection of the image patches, while the output matrix performs a categorical projection.
> - We note that the semantic task learned by these networks is different: Pre-trained LLMs are trained to predict the next word of a sentence (only auto-regressor LLMs were investigated in similar previous NLP studies), while ViTs for image classifications are trained to predict the semantic label of an image.
> - We discuss the implications of the above differences and how they may lead to differences in the information that is encoded in key-value memory pair systems. Particularly, we give a more thorough explanation as to why interpreting the keys of ViTs is not as straightforward as in the LLM case: while the input space of LLMs continues to be relevant throughout the network’s hierarchy because it is also the space that is projected to in the output prediction (see [1]), in ViTs the input-space is no longer relevant for later projections. That is why our current work focuses on analyzing the semantics of this system’s value vectors. We however note that future work can be dedicated to finding alternative ways of investigating what is encoded in the keys of these systems.
>
> _Weakness 2_
>
> We agree on the value of investigating the generalizability of our framework and findings by probing ViTs trained on other datasets. We have now done so more thoroughly. As described in detail in the general response, in the new version of our paper we have expanded our analyses to a ViT-B/16 fine-tuned on CIFAR100, and a ViT-B/16 trained on a higher-quality and a multi-labeled version of ImageNet-21k (MIIL) [2].
> We found that our framework can be used to analyze both variants of ViT, by successfully translating their [CLS] and image tokens across layers into the class-embedding space to get meaningful insights into how they build categorical representations and the effect of the training dataset in these processes.
>
> Concretely, we found that fine-tuning on CIFAR100 increases the identifiability score of tokens, from the very first layers of the model (see Fig. 1 in rebuttal PDF). This result suggests that fine-tuning ViTs in small datasets creates a class-embedding representation that is potentially overfitted to detect simple patterns in the image. Training in MIIL improves the identifiability of tokens (especially the [CLS]) without showing overfitting patterns, compared with a vanilla ViT-B/16 (see Fig. 1 in rebuttal PDF).
> Fig. 2 in the rebuttal PDF also shows how both variants rely on key-value memory pair mechanisms, with MIIL showing the strongest key-value agreement rates of all networks, meaning that more tokens promote categories using these mechanisms than other ViT variants.
> We have included these experiments and interesting findings in the new version of our manuscript.
>
> We thank you again for the concrete suggestions that we believe to have sufficiently addressed. If so, please consider raising the reviewer rating.
>
> [1] Dar et al. “Analyzing Transformers in Embedding Space.” (2023).
> [1] Ridnik et al. "Imagenet-21k pretraining for the masses." (2021).

---

> > ### Comment · Reviewer_mxXe · 2023-08-18
> > **Post rebuttal**
> >
> > Thank you for spending the time addressing my initial queries with the paper. Your inclusion of the discussions on the differences from similar work in NLP and an investigation on the generalizability of the proposed framework is appreciated. Kindly ensure that these discussions are seamlessly integrated into the revision.
> >
> > As of now, I would like to remain my score. Looking forward to the opinions of the other reviewers.

---

### Official Review · Reviewer_5aAZ · 2023-07-07

**Soundness:** 3 good
**Presentation:** 3 good
**Contribution:** 2 fair
**Rating:** 5
**Confidence:** 3

**Summary:**

This paper utilizes a pre-trained embedding matrix to elucidate the mechanism of Vision Transformers. By using the embedding matrix, inner representations at any layer can be investigated in class spaces. Specifically, it offers a visualization of how self-attention and MLP process class categorical information. This method can also be employed to visualize the saliency map.

**Strengths:**

1. Applying the embedding matrix is straightforward. This method does not necessitate additional training and is easy to implement.
2. Overall, the paper is well-written and well-organized

**Weaknesses:**

1. It appears that the main point of this paper is introducing the embedding matrix for empirical analysis in vision tasks and demonstrating the usefulness, since the embedding matrix has been previously explored in NLP tasks. However, the novelty and significance of the insights obtained by using the method are somewhat limited.
    - Non-zero linear probing accuracy can be achieved even at the early layers [a].
    - Self-attentions significantly change the representations, compared to MLPs [a].
    - Self-attentions and MLPs perform complementary roles. For example, self-attentions aggregate information whereas MLPs diversify it [b].
2. Analyses are provided only for vanilla ViT. Since modern ViTs such as the Swin Transformer utilize global average pooling instead of a CLS token, the influence of these findings might be limited.

[a] Raghu, Maithra, et al. "Do vision transformers see like convolutional neural networks?." *NeurIPS* (2021).

[b] Park, Namuk, and Songkuk Kim. "How do vision transformers work?." ICLR (2022).

**Questions:**

A straightforward, and potentially more accurate, method to generate a map from token space to class space involves conducting layer-wise linear probing experiments (see, e.g., Figure 13 of [a]). When compared to linear probing, does the pre-trained embedding matrix offer any advantages? One potential advantage I can foresee is that no additional learning is required. Incorporating a comparison between linear probing and the embedding matrix might improve the manuscript.

**Limitations:**

See the Weaknesses section for technical limitations. I cannot find any ethical issue.

---

> ### Author Rebuttal · Authors · 2023-08-09
>
> We thank the reviewer for their valuable feedback, which has helped clarify and better demonstrate the novelty, generalizability, and usefulness of our framework and findings.
>
>
> _Weakness 1_:
>
> Thank you for sharing relevant previous work which we now cite and discuss in the revised manuscript. We note that our work provides additional and complementary insights to those mentioned in the reviewer's comment. Specifically:
> - [a] shows that non-zero linear probing can be achieved in the [CLS] token, but as can be seen in Fig. 11 of their work, their linear probes did not achieve good accuracy when investigating the image tokens. In contrast, our method can successfully extract categorical information from image tokens and characterizes the factors enabling the emergence of class representations (e.g. the categorical construction in class-labeled vs context-labeled tokens). Moreover, as our general rebuttal and the “linear probing” section in this response elaborates in detail, linear probing experiments do not enable the same inferences as our method does.
> - Using a Ratio of Norms analysis, [a] shows that the self-attention layers impact the residual stream more strongly than MLP layers in the [CLS] token at early layers, and the reverse is true for image tokens at later layers (MLP more than self-attention layers). Importantly, this Ratio of Norms analysis only provides an estimate of the proportion of information added to the residual stream by these layers, in comparison with that added by the skip connections. It does not characterize the amount and the mechanisms by which the information added by each layer and block updates the categorical representations of the residual, as we did in our work. The mechanistic insights that we have gained with our approach can potentially be used for model editing or performance improvements in future work.
> - The results of [b] are indeed relevant to our work and are now cited in the revised manuscript. However, our findings are complementary to those reported in [b]. Similarly to [a], [b] analyzes statistical properties of the information added by self-attention and MLP layers of ViTs. In contrast, our work aims to provide insights into the semantic characteristics of the information added by these layers. Concretely, we investigate how categorical representations are built by exploiting mechanisms like key-value memory pairs that encode semantic information, which have been reported in NLP models.
>
> More broadly, the greatest difference between our work and those mentioned in the reviewer's comment is that we introduce a general mechanistic interpretability method that can be efficiently applied to probe the categorical representations of different and future types of ViTs, as long as a class-embedding matrix has been learned during training.
>
> In the new version of our manuscript, we explicitly clarify the novelty and significance in the “Related Work” section; specifically, how our findings provide additional insights to those mentioned in the review.
>
>
> _Weakness 2_:
>
> Thank you for pointing out this limitation, which we now overcome with additional experiments yielding favorable results. As described in detail in the general comment, we have expanded our analyses to other variants of ViTs, including those using GAP instead of the [CLS] token. This new set of experiments demonstrates that we can successfully apply our framework to other types of ViTs to gain insights into how different training variables (e.g. dataset, architecture, pooling, learning objective), affects category building mechanisms.
> For example, we found that introducing GAP does not decrease the class identifiability scores of image tokens across layers (see Fig. 1 of rebuttal PDF), but it does decrease the class identifiability rate of the tokens in the last layer (i.e. the percentage of tokens that contain a class identifiability score of 1). In addition, as Fig. 2 and 3 of the PDF show, GAP training decreases the reliance on key-value memory pair mechanisms for building categorical representations. Taken together, these findings corroborate the idea that the categorical representations on GAP-based ViTs emerge in a more distributed fashion than in [CLS]-based ViTs. However, we also found that class and context tokens have significant differences in the evolution of their identifiability scores, which allows us to conclude that even in these networks there is a meaningful pattern of how identifiability emerges from the image tokens.
> In summary, these new experiments show that our framework can be used to investigate modern GAP-based ViTs, which strengthens the generalizability of our work.
>
>
> _Question_:
>
> Thank you for raising this point and for suggesting linear probing experiments. We have run additional experiments accordingly, with informative and favorable outcomes. Please see the general response where we describe how we empirically demonstrate the advantages of our method over linear probing.
>
> Here, we also want to add that the linear probe experiment reported in Figure 13 of [a] aggregates the representations of the tokens. Thus, this approach does not take into account the potential differences in class identifiability between image and [CLS] tokens, or between different kinds of image tokens (e.g. class-labeled vs context-labeled image tokens, as explored in our study). For that reason, we instead replicated the experiments of Figure 9 of [a]. Moreover, as described in the general comment, we obtained different results than those reported in Figure 9 of [a], which may be taken as evidence of a lack of generalizability across training conditions in linear probing experiments.
>
> We benefited from the comments and suggestions, and believe we have addressed them successfully. If so, please consider raising the reviewer rating.

---

> > ### Comment · Reviewer_5aAZ · 2023-08-16
> > **RE: Rebuttal by Authors**
> >
> > I appreciate the author's effort to further generalize the discussion and clarify its novelty. However, I still believe that the novelty is limited, and I am not fully convinced by the discussion on linear probing based on ad hoc analysis. Moreover, as reviewer vRFW pointed out, it might not be clear whether the representations are aligned. Nevertheless, as the paper has improved during this rebuttal process, I am inclined to change my recommendation to borderline accept.

---

> > > ### Author Response · Authors · 2023-08-16
> > >
> > > Thank you for your comments and for raising the rating. We respond to each in turn.
> > >
> > > > However, I still believe that the novelty is limited,
> > >
> > > To improve the quality of our work, we would appreciate if you could further clarify why the broad and novel contributions of our work (outlined in the rebuttal) over the studies mentioned in the review are insufficient.
> > >
> > > > and I am not fully convinced by the discussion on linear probing based on ad hoc analysis.
> > >
> > > Similarly, we would appreciate it if you could clarify the meaning of “based on ad hoc analysis”.
> > >
> > > In response to the original reviewers’ comments on linear probing, we have replicated the suggested linear probe analysis of [a] and compared it with our method, reporting favorable results. Specifically, we believe our results give strong evidence that linear probing does not necessarily uncover the information driving the alignment to the class-embedding representation used in the categorical prediction, as our method does (please see rebuttal for details). Moreover, in contrast to our method, it is not possible to use linear probes to investigate the mechanisms by which categorical alignment takes place.
> > >
> > > We would appreciate concrete, actionable feedback on how to strengthen our conclusions on this point.
> > >
> > > > Moreover, as reviewer vRFW pointed out, it might not be clear whether the representations are aligned.
> > >
> > > As discussed in the response to reviewer vRFW, whether representations are aligned in absolute terms or not is orthogonal to our stated research goals. We do not claim that early layers lack information because they are not aligned to class-embedding representations.
> > >
> > > Instead, our approach aims to characterize how the alignment to the class-embedding space unfolds throughout the model’s hierarchy and the mechanisms that enable it. Thus, our approach characterizes relative alignment change by means of an identifiability measure that is continuous (as opposed to discrete measures of previous work).
> > >
> > > Moreover, we show that the alignment process takes place from very early layers (as evidenced by the significantly increased alignment of inner representations to the class embedding as compared to a random model), so we do not disregard the importance of these early (and less aligned in absolute terms) representations with our method.

---

### Author Rebuttal · Authors · 2023-08-09

We thank the reviewers for feedback that helped improve our manuscript. In the general response, we address two common concerns about our work: we (1) report new experiments that provide evidence of the generalizability of our method, and (2) conceptually discuss and empirically demonstrate the advantages of our framework over linear probes. In all cases, we obtain results that are informative and favor our proposed method.

1) Generalizability

Some reviewers raised the possibility that our framework might not generalize outside vanilla ViTs. We agree that more experiments are needed to demonstrate generalizability. We have thus tested 5 additional variants of ViTs and obtained favorable results in all cases.

Specifically, we separately probed ViTs that are 1) larger (ViT-L/16); 2) fine-tuned for other datasets (CIFAR100); 3) trained on higher-quality datasets (MIIL) [1]; 4) trained with a refinement module that aligns all tokens’ intermediate representations to class space [2]; and 5) trained with Global Average Pooling (GAP) instead of the [CLS] token.

Our findings demonstrate that we can successfully translate the [CLS] and image tokens of all these variants into the class-embedding space across layers. As depicted in Table 1 (see rebuttal PDF), the identifiability rate of the image tokens in the last layer was above chance in all models. In addition, Fig. 1 shows class identifiability increases over blocks for all variants. We also found that in all networks, class-labeled tokens have higher identifiability scores than context-labeled ones, which illustrates that our method can be used to characterize the emergence of categorical representations in image tokens.

Our results also prove that we can use our framework to inspect the role of attention and MLP layers across ViT variants, and their use of key-value memory pair systems (Fig. 2). Although we observe similar patterns across networks, our method also reveals quantifiable differences indicating how the modification of vanilla ViTs affect the building of categorical representations. We discuss these differences in the responses to each reviewer comment.

We have added the above findings to our paper and: (1) swapped Fig. 2a and 4 (main paper) with Fig. 3 and 4 (PDF), (2) discussed the similarities and differences across ViT variants, and (3) included figures for each network in the supp. material.

2) Comparison to linear probing

Some reviewers point out that we do not discuss the advantages of our method over the commonly used linear probing approach. In the revised manuscript we perform experiments to demonstrate that linear probing is less informative for our research question.

We first note that these two methods differ in the type of insights they can provide. Although both aim to characterize the information encoded in the hidden representations, only our framework can quantify how these intermediate representations increasingly align with the class prototype learned by the model (encoded by the weights of the class-embedding matrix). Furthermore, our method additionally characterizes the inner mechanisms of this alignment (e.g. by investigating key-value memory pair systems via the projection of ViT’s parameter matrices).
In contrast, linear probes measure the linear separability in the hidden representations of samples of different classes [3] but do not reveal whether the learned separability is exploited by the model in categorical decisions. Thus, linear probing does not necessarily uncover the relevant factors behind category-building processes in ViTs. In other words, the successful class probing of tokens in a given layer can rather reflect the separability of confounders whose feature representations are ignored by the model in the class-embedding space. For example, for a particular token and layer, a linear probe may use internal representations of background information that is highly correlated with the presence of a class in a specific dataset, to distinguish between categories. We show this is likely through the experiments described next.

Following [12], in the revised paper we trained separate 10-shot linear classifiers on ImageNet for each token position and layer of a VIT-B/32. To test if the information learned by these probes shed light on the categorical decisions taken by the network, we conducted negative and positive perturbation tests. Concretely, we quantified the class identifiability scores obtained from the linear probes for each token, gradually removed the tokens with the least to most identifiable scores (for negative perturbation; vice-versa for positive perturbation), and measured the accuracy of the model with each removal.

Our results demonstrate that even if linear probes can generally decode with better top-1 accuracy the classes of the image tokens, this is not informative of the relevance of each token for the categorical decision of the network (Fig. 3). Moreover, we found that our experiments achieve higher accuracy than those reported in [12], which evidences the lack of generalizability of linear probes across training conditions (e.g. different datasets).

Finally, note that even if linear probes and our method would provide similar insights (which we demonstrated is not the case), our framework is more time-efficient: it comprises a one-forward pass on validation images, and zero-pass when projecting the parameter matrices. In contrast, linear probes additionally involve 1) a one-forward pass over the training images, and 2) the fitting of a linear classifier for every token position and layer.

We add these linear probe results to a new subsection in the main manuscript, and Fig. 3 (PDF) to the supp. material.

[1] Ridnik et al. "Imagenet-21k pretraining for the masses" (2021);
[2] Naseer et al. “On Improving Adversarial Transferability of Vision Transformers” (2021);
[3] Alain & Bengio. "Understanding intermediate layers using linear classifier probes" (2016);

---

### Decision · Program_Chairs · 2023-09-21

**Decision:**

Accept (poster)

**Comment:**

The paper performs an analysis of trained Vision Transformers. In particular, it proposes an analysis technique that uses the pre-trained network head as a probe to quantify class-alignment in different parts of the network. The paper received mixed reviews, that lean positive on average. Concerns include: the lack of actionable conclusions from the analysis, value over the standard linear probing technique, and generality of the results to other ViT flavours and datasets.

In the rebuttal the authors present a compelling experimental evidence that their class-based probe has substantial orthogonal value to linear probing, plus they provide analysis on additional networks and datasets. These address the second two concerns. Regarding "actionable conclusions", I believe that thoughtful analysis that provides new insights, such as those presented here, can still be a valuable contribution to the community, even if the take-aways for future network design/training are not immediately obvious. I do not see any blocking concerns remaining after the rebuttal, therefore I recommend acceptance.